# Constitutively active Lyn kinase causes a cutaneous small vessel vasculitis and liver fibrosis syndrome

Neutrophilic inflammation is a hallmark of many monogenic autoinflammatory diseases; pathomechanisms that regulate extravasation of damaging immune cells into surrounding tissues are poorly understood. Here we identified three unrelated boys with perinatal-onset of neutrophilic cutaneous small vessel vasculitis and systemic inflammation. Two patients developed liver fibrosis in their first year of life. Next-generation sequencing identified two de novo truncating variants in the Src-family tyrosine kinase, *LYN*, p.Y508*, p.Q507* and a de novo missense variant, p.Y508F, that result in constitutive activation of Lyn kinase. Functional studies revealed increased expression of ICAM-1 on induced patient-derived endothelial cells (iECs) and of β2-integrins on patient neutrophils that increase neutrophil adhesion and vascular transendothelial migration (TEM). Treatment with TNF inhibition improved systemic inflammation; and liver fibrosis resolved on treatment with the Src kinase inhibitor dasatinib. Our findings reveal a critical role for Lyn kinase in modulating inflammatory signals, regulating microvascular permeability and neutrophil recruitment, and in promoting hepatic fibrosis.

Next-generation sequencing (NGS) of disease probands and their parents has propelled the discovery of monogenic causes of autoinflammatory diseases[1], that present with severe, perinatal-onset, systemic, and organ-specific inflammation. Tissue infiltration by activated innate immune cells, including monocytes, macrophages, and neutrophils, if untreated, leads to rapidly progressive organ injury, that can involve many organ systems. The discovery of de novo gain-of-function (GOF) mutations in IL-1 activating inflammasomes (NLRP3, NLRC4, pyrin, and NLRP1) and Type-1 IFN activating viral sensors (STING, RIG-I, and IFIH1/MDA5) linked key innate immune pathways to sterile inflammation. Repurposing drugs that target inflammatory cytokines has confirmed the prominent roles of pro-inflammatory cytokines, including IL-1, Type-I IFN, and TNF in the disease pathogenesis of autoinflammatory diseases, and identified the inflammasome and viral sensors as novel targets for drug development[2].

A role of Src family kinase members in causing sterile inflammation and human disease is only recently recognized. The often redundant roles of Src kinases in modulating surface receptor signaling including TLRs, Fc receptors, integrins, growth factor receptors, and adhesion molecules[3,4], and their role in regulating complex cellular functions such as proliferation, cell differentiation, apoptosis, migration, and metabolism have been well documented[5]. With the discovery of *SRC* as the first human proto-oncogene[6], and the recognition that somatic GOF mutations in *SRC* promote colon cancer and liver metastasis[7], and that germline GOF *SRC* mutations cause thrombocytopenia, myelofibrosis, and non-inflammatory bone pathologies[8], an oncogenic potential of other Src family kinase members has been of concern. Surprisingly, GOF mutations in the Src family kinase members *FGR*[9] and *HCK*[10] that, in contrast to *SRC*, are all highly expressed in human neutrophils and monocytes, have been reported to cause systemic inflammatory diseases, which has renewed interest in targeting specific Src kinases in inflammatory diseases[11]. Here, we describe three unrelated boys with three de novo mutations in the Src-family tyrosine kinase, Lyn kinase, *LYN;* all presented with systemic inflammation and recurrent neutrophilic small vessel vasculitis. Two patients with truncating mutations had liver

✉ e-mail: goldbacr@mail.nih.gov

fibrosis that in one patient resolved on treatment with the Src kinase inhibitor dasatinib. We characterize the role of increased Lyn kinase activity in neutrophils, endothelial cells and lesional liver biopsies and utilized an iPSC-derived endothelial cell platform for disease modeling of neutrophilic vasculitis and to screen and evaluate drug efficacy.

## Results

### Identification of the *LYN* mutations and clinical characterization

We performed trio whole-exome sequencing (WES) on whole blood from Patient 1, who presented with unexplained systemic inflammation, and his parents, and identified a nonsense germline mutation in a coding region of the Src kinase, *LYN*, c.1524C>G, p.Y508* (transcript NM_002350), that resulted in truncation of five terminal amino acids, including a regulatory tyrosine at position p.Y508. In Patient 2, a targeted NGS panel revealed a missense mutation in *LYN* c.1523A>T, resulting in the replacement of tyrosine at position 508 by a phenylalanine (p.Y508F); and clinical WES in Patient 3 identified a nonsense germline mutation in *LYN*, c.1519C>T, p.Q507* resulting in loss of six terminal amino acids including p.Y508. The three mutations occurred de novo (Fig. 1a, b and Supplementary Fig. 1), and were predicted to be deleterious. All were absent in public databases including gnomAD. *LYN* variant features, including amino acid conservation and pathogenicity predictions are detailed in Supplementary Table 1.

Clinical and laboratory features of the three patients are outlined in Table 1. All three patients developed diffuse purpuric skin lesions (Fig. 2a, b), fever, and increased C-reactive protein (CRP) within the first hours of life. Patient 1 and Patient 3 had hepatosplenomegaly, transaminitis, and severe thrombocytopenia. Patient 1 underwent a splenectomy for thrombocytopenia and developed leukocytosis, thrombocytosis, and persistent anemia post-splenectomy. At the age of 21 months, he also had transiently elevated circulating auto-antibodies including ANA, anti-Sm, anti-SSA, anti-mitochondrial, and anti-phospholipid antibodies without clinical autoimmune manifestations. Persistently elevated liver function tests (LFTs) led to a liver biopsy at 22 months, which showed a mild periportal lymphocytic infiltrate, clusters of macrophages and scattered neutrophils within hepatic sinuses and portal areas, marked biliary ductopenia and early peri-sinusoidal fibrosis (Fig. 2c). Other clinical manifestations included intermittent abdominal and testicular pain, headaches, conjunctival and periorbital erythema (Supplementary Fig. 2a), arthralgias, and myalgias worse after exercise or trauma. A magnetic resonance imaging (MRI) during myalgia showed fasciitis and muscle edema (Fig. 2a). Patient 3 was recently born at 37 weeks of gestational age with intrauterine growth restriction (IUGR) and congenital hydrocele. At the age of 5 months, liver elastography was consistent with liver cirrhosis. His thrombocytopenia improved on etanercept and initiation of dasatinib is considered if transaminitis and high elastography scores do not improve.

a    Patient Pedigrees

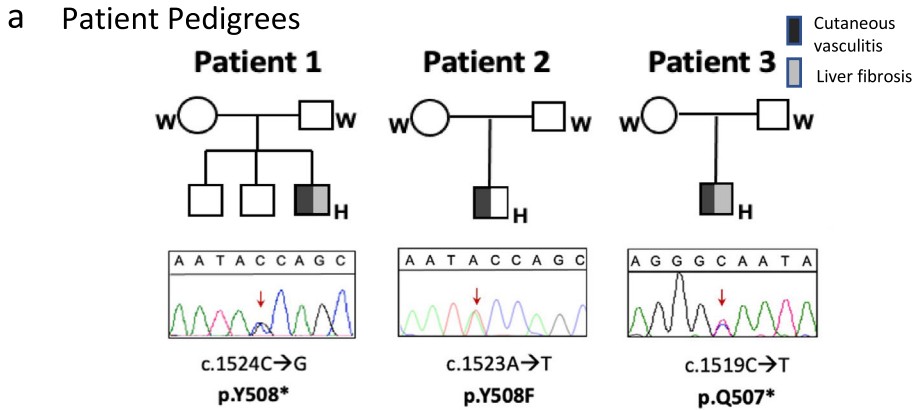

b    *LYN* mutations and chromosomal organization

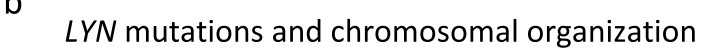

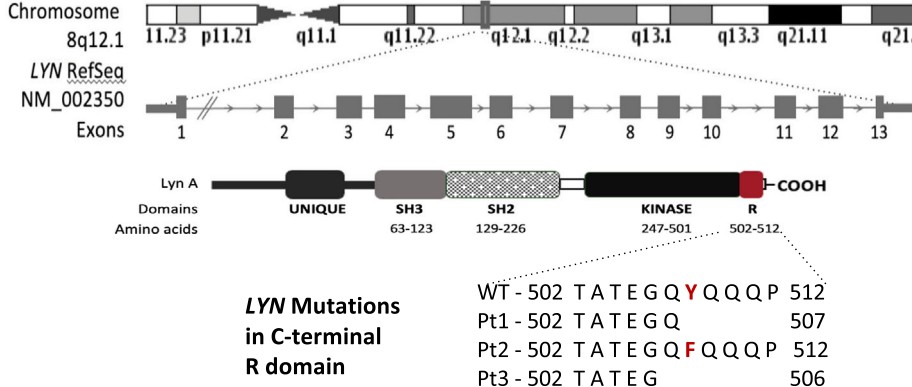

**Fig. 1 | Discovery of de novo GOF mutations in *LYN* as cause of cutaneous vasculitis and liver fibrosis syndrome. a** The pedigrees show a de novo variant in *LYN* in each of the three patients. Squares and circles represent male and female family members, respectively; solid symbols and open symbols indicate affected and unaffected family members, respectively. **b** In Patient 1 and Patient 3 the truncating mutations p.Y508* and p.Q507* result in the loss of the 5 or 6 terminal amino acids, respectively; and in Patient 2, the amino acid substitution from a tyrosine to a phenylalanine prevents phosphorylation in the C-terminal regulatory domain of Lyn kinase.

**Table 1 | Clinical features in patients with gain-of-function mutations in *LYN***

| Clinical and laboratory features | Patient 1 | Patient 2 | Patient 3 |
|---|---|---|---|
| **Clinical presentation** | | | |
| Age of disease onset | 1st day of life | 1st day of life | 1st day of life |
| Age at diagnosis | 2 years 6 months old | 15 years-old | 4 months old |
| Presenting Symptoms | Purpuric rash, hepatosplenomegaly, fever, thrombocytopenia at birth | Mild purpuric rash at birth, fever, and generalized severe purpuric rash at the age of 3 months | Hepatosplenomegaly, thrombocytopenia, and discrete purpuric rash at birth |
| Hydrops fetalis | Yes, intra-utero platelet and PRBC transfusion at 29 weeks of GA | No | No, had congenital hydrocele |
| Liver fibrosis | Yes | No | Yes |
| Other clinical manifestations | Recurrent parotitis, abdominal pain, periorbital edema and erythema, conjunctivitis, epididymitis, headaches, arthralgias, oral ulcers, fatigue, GVHD-like colitis | Recurrent abdominal pain, periorbital edema and erythema, conjunctivitis, epididymitis, headaches, arthralgias, oral ulcers, fatigue, GVHD-like colitis | Intrauterine growth restriction, failure to thrive, transient periorbital erythema, jaundice, direct hyperbilirubinemia |
| **Laboratory findings[a]** | | | |
| ESR (mm/1h) | 64 | ND | ND |
| CRP (mg/L) | 14–86.5 | 46–166 | 6.5–107.6 |
| SAA (mg/L) | ND | 182–984 | ND |
| CBC | Mild anemia, mild leukocytosis, moderate to severe thrombocytopenia | Mild leukocytosis | Mild anemia, moderate leukocytosis, moderate to severe thrombocytopenia |
| LFTs | Increased ALT, AST and GGT | Normal ALT, AST and GGT | Increased ALT, AST and GGT |
| Autoantibodies | Positive ANA, anti-Sm, anti-SSA, ACL IgG, LAC, anti-mitochondrial, RF, anti-TPO | Negative by clinical tests, transient positivity for ANA on research testing once | Borderline anti-PR3 |
| Complement | nl CH50, C3, C4 | nl C3, C4 | ND |
| CD4/CD8/B/NK lymphocytes (abs #) | High/high/nl/nl | nl/nl/nl/nl | High/high/high/high |
| IgG/IgA/IgM | nl/nl/low | nl/nl/low | nl/low/nl |
| Skin biopsies | Small vessel vasculitis with neutrophilic infiltrate and destruction of dermal vessel walls | Perivascular neutrophilic dermal infiltrate | Small vessel vasculitis with neutrophilic infiltrate and destruction of dermal vessel walls |
| **Liver evaluation** | | | |
| Elastography (max) | 6.7 kPa | ND, normal LFTs | 18.7 kPa |
| Infections | Enteropathogenic E. coli, Salmonella sp, Toxocara canis[b] | Post-streptococcal glomerulonephritis | Late neonatal sepsis, COVID-19 (asymptomatic) |
| Treatment | Poor response to IVIG, IVMP and oral prednisolone, partial response to dasatinib monotherapy, partial response to etanercept monotherapy, good response to dasatinib and etanercept combination therapy | No response to anakinra and tocilizumab, partial response to colchicine and good response to etanercept and colchicine | Partial improvement of CRP and thrombocytopenia, resolution of direct hyperbilirubinemia, return of normal growth rate, and persistence of liver fibrosis on etanercept therapy |

*PRBC* packed red blood cells, *GA* gestational age, *GVHD* graft versus host disease, *ESR* erythrocyte sedimentation rate, *CRP* C-reactive protein, *SAA* serum amyloid A, *CBC* complete blood count, *LFT* liver function tests, *ALT* alanine aminotransferase, *AST* aspartate aminotransferase, *GGT* gamma-glutamyltransferase, *ANA* anti-nuclear antibody, *Sm* Smith, *SSA* Sjogren's syndrome A, *ACL* anti-cardiolipin, *LAC* lupus anticoagulant, *RF* rheumatoid factor, *TPO* thyroid peroxidase, *PR3* proteinase 3, *Igs* immunoglobulins, *abs #* absolute number, *max* maximum value detected, *IVIG* intravenous immunoglobulin, *IVMP* intravenous methylprednisolone, *kPa* kilopascal, *nl* normal, *ND* not done.
For definition of poor response, partial response, and good response, please see the online Supplementary Information file.
[a]Depicted laboratory findings were collected pre-treatment.
[b]All infections occurred on dasatinib or dasatinib + etanercept therapies.

## Histopathology of vasculitic skin lesions, liver, and colon biopsies

All patients had histopathological evaluation of lesional skin biopsies, which showed a dense neutrophilic infiltrate around small vessels including capillaries (Fig. 2a, b and Supplementary Fig. 2b, c) suggestive of pauci-immune small vessel vasculitis (ANCA-negative). Extravasated neutrophils formed neutrophil extracellular traps (NETs) in areas of loss of vessel wall integrity (Fig. 2b and Supplementary Fig. 2c). Lyn kinase is regulated similarly to other Src family kinases (Supplementary Fig. 3a) and it is expressed in endothelial cells of small vessels, in neutrophils, monocytes, and macrophages (Fig. 2b, Supplementary Figs. 2b and 3b–d) and in liver sinusoidal endothelial cells (LSECs) (Fig. 2c). Patient 1 and Patient 2 developed both post-infectious diarrhea that resolved without treatment; colon biopsies showed apoptotic crypt injury, reminiscent of graft-versus-host disease (Fig. 2d).

## Phosphorylation assays in wild-type and mutant Lyn

To further investigate the functional impact of the de novo mutations, we assessed Lyn kinase phosphorylation. The C-terminal tail tyrosine residue, p.Y508, inactivates wild-type Lyn kinase by association with its own SH2 domain (Fig. 1b, Fig. 3a, left panel)[12,13]. We hypothesized that the C-terminal deletions of Lyn kinase p.Y508 or the p.Y508F substitution increase Lyn kinase activity (Fig. 3a, left panel)[5]. Transfection of mutant Lyn into HEK293FT cells showed constitutive phosphorylation of the kinase activating tyrosine, p.Y397, and absent phosphorylation of the "inhibitory tyrosine", p.Y508 (Fig. 3a, right panel). Mutant Lyn kinase caused increased phosphorylation of several Lyn kinase targets including the adapters, Scimp, involved in MHC-II signaling transduction in B cells[14], in sustaining CLEC7A/DECTIN1 signaling in dendritic cells[15], and in scaffolding Lyn to activated TLRs[16], and Skap2, that is involved in β2 integrin activation and neutrophil recruitment (Fig. 3b)[17]. The broad

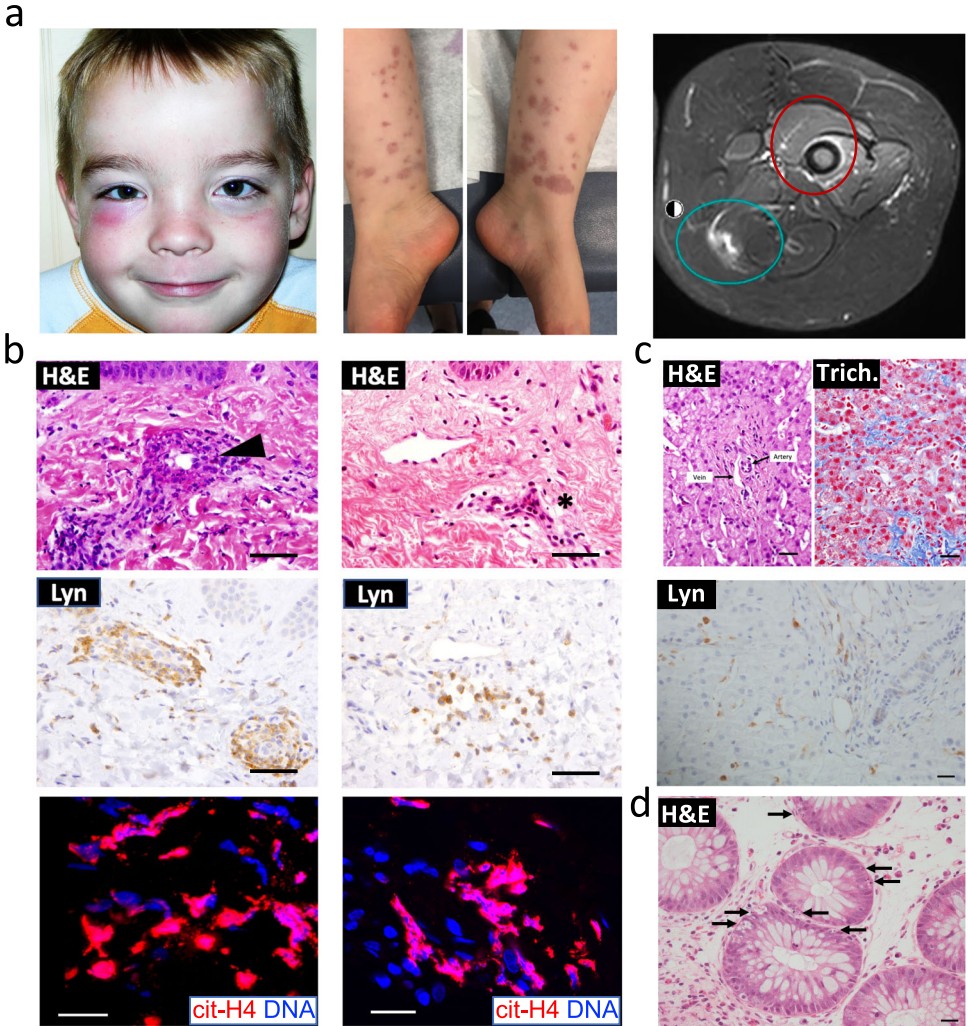

**Fig. 2 | Clinical and histopathologic features of patients with *LYN* gain-of-function mutations. a** Periorbital edema and erythema in Pt.2 (left panel), non-blanching vascular rasciis in Pt.1's lower extremities (center panel), and left thigh MRI from Pt.1 (right panel) depicting fasciitis (green circle) and periostitis (red circle) are shown. **b** Hematoxylin & eosin (H&E) staining from lesional skin biopsies of Pt.1 (first column images) and Pt.2 (second column images) show destruction of a small vessel with surrounding inflammatory cells in Pt.1 (black arrow) and a peri-vascular infiltrate in Pt.2 (asterisk) (upper panels, scale bars, 100 μm). The endo-thelial cells and surrounding neutrophils are Lyn kinase (Lyn) positive in both patients (middle panels, scale bars, 100 μm). The lower panels show the localized accumulation of neutrophil extracellular traps (NETs) (pink) surrounding the small vessels in Pt.1 and Pt.2, respectively. cit-H4, citrulline histone H4, scale bars, 20 μm. **c** A Liver biopsy performed in Pt.1 at the age of 22 months illustrates a portal area lacking a bile duct on H&E (left upper panel) and early peri-sinusoidal fibrosis on Masson's trichrome staining (right upper panel). Lyn staining of sinusoidal cells and endothelial cells in the portal areas is shown in Pt.1's liver biopsy that was per-formed at the age of 4 years (lower panel). Scale bars, 20 μm. **d** Colon biopsy from Pt.2 shows apoptosis of multiple crypt epithelial cells (black arrows). Scale bar, 20 μm.

phosphorylation of mutant Lyn kinase is also shown for SAM68 and a broad spectrum of phosphorylated downstream targets (Supplementary Fig. 4 and 5). Flow cytometry analysis of anti-IgM stimulated B cells from Patient 1 showed increased phosphorylation of Lyn kinase substrates, including PLCγ2, CD19, CD79A that was blocked by the Src kinase inhibitors dasatinib and PP2 (Supplementary Fig. 5). Furthermore, B cell stimulation assays reveal defects in central and peripheral B cell tolerance (see Supplementary Figs. 11–13 in the Supplementary Results).

### Evaluation of inflammatory response

Serologic analysis from untreated Patients 1, 2, and 3 showed elevated CRP and pro-inflammatory cytokines (i.e., IL-6) (Fig. 3c, left panel and Fig. 4a). Patients 1 and 3 had marked elevation of biomarkers of neu-trophil (sL-selectin and lipocalin) (center panel) and endothelial cell (i.e., sICAM-1 and sE-selectin) (right panel) activation, comparable to patients with NOMID and SAVI (Fig. 3c, middle and right panel).

Monocyte-derived macrophages, upon LPS stimulation, released increased levels of cytokines TNF-α and IL-6, and chemokines CXCL10 and CCL3/4/5, and shed increased levels of sICAM-1 when compared to healthy and disease controls (Supplementary Fig. 6). Neutrophils showed signs of constitutive activation, including low surface expression of CD62L (Supplementary Fig. 7) and high expression of β2-integrin adhesion molecules (Fig. 4b), with normal cytokine responses to a broad range of endogenous, Toll-like receptor (TLR) and microbial stimulants (Supplementary Fig. 7 and Supplementary Table 2). In lesional skin biopsies, ICAM-1 was upregulated and colocalized with the endothelial marker CD31/PECAM along the endothelial wall (Sup-plementary Fig. 2c).

### Clinical response to targeted treatment with dasatinib and TNF inhibition

All patients responded partially (Patient 1 and Patient 3) or fully (Patient 2) to treatment with the TNF inhibitor etanercept after failing

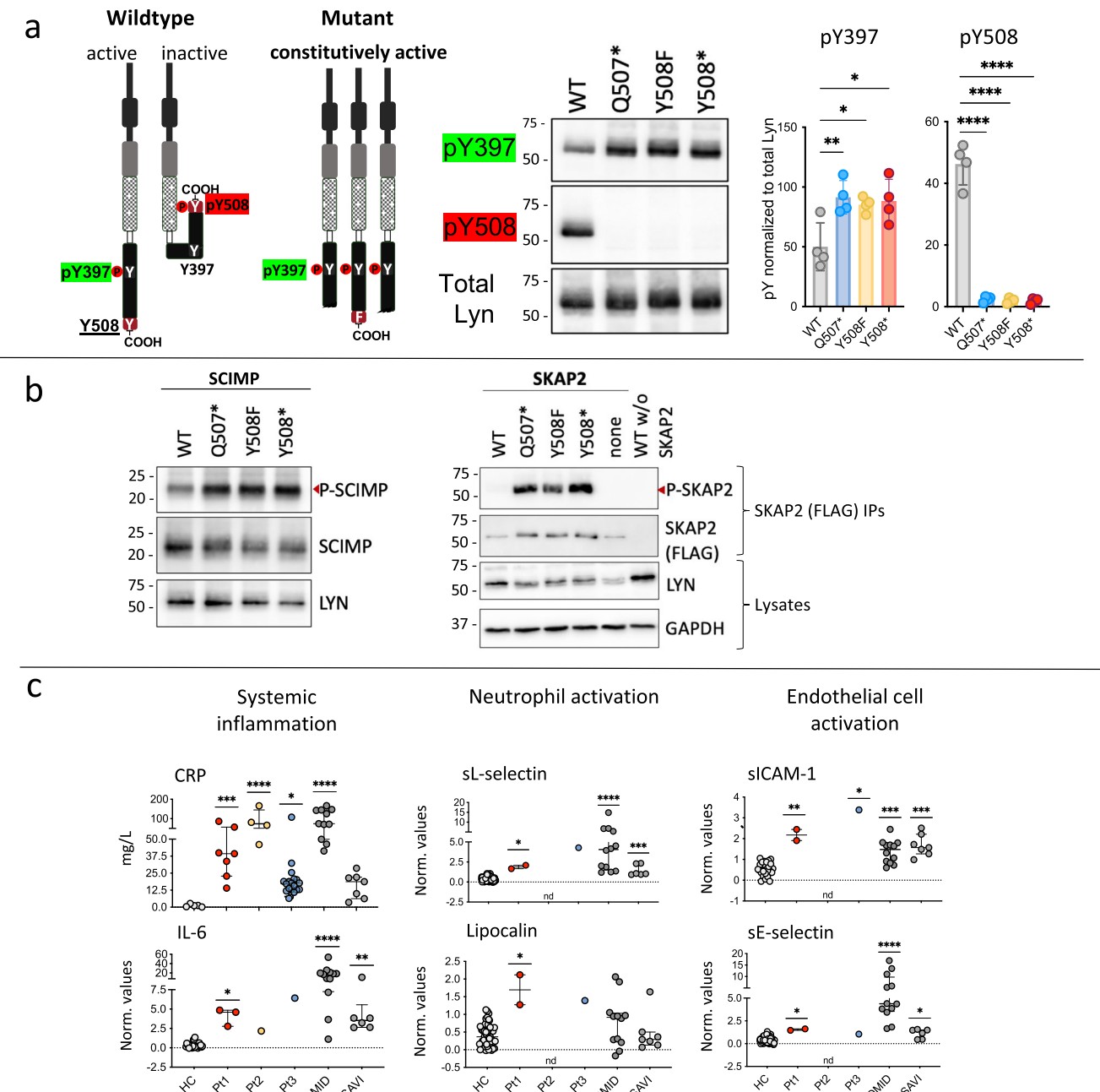

**Fig. 3 | Constitutive Lyn kinase activation and evidence of systemic inflammation and endothelial activation. a** Phosphorylation-dependent active and inactive configuration of Lyn kinase. Disease-causing *LYN* variants lead to constitutive activation (left panel). Western blot of lysates from transiently transfected HEK293FT (wild-type (WT) or mutant *LYN*) demonstrate increased phosphorylation of p.Y397 and absent phosphorylation of p.Y508 in the mutant constructs (center panel). Data of 4 biologically independent experiments is graphed in a scatter plot with bar graphs depicting mean values ± SD. p.Y397 comparisons, WT versus: p.Q507*, $p = 0.0082$, p.Y508F, $p = 0.0228$; or p.Y508*, $p = 0.0137$. p.Y508 comparisons, WT versus: p.Q507*, $p < 0.0001$, p.Y508F, $p < 0.0001$ or p.Y508*, $p < 0.0001$. *$p < 0.05$, **$p < 0.01$, ****$p < 0.0001$ as determined by ordinary one-way ANOVA with Bonferroni's multiple comparison post-test. **b** Phosphorylation of Lyn adaptor proteins, Scimp (left panel), and Skap2 (right panel) in transiently *LYN* construct co-transfected HEK293FT cells. Red arrow heads indicate phosphorylated Lyn kinase substrates. Depicted are representative images of n = 3 independent experiments for Scimp and Skap2. **c** Pre-treatment samples from the 3 patients (Pts.1-3) and from patients with the systemic autoinflammatory diseases, neonatal-onset multisystem

inflammatory disease (NOMID), or STING-associated vasculopathy with onset in infancy (SAVI), were compared with healthy controls (HC). For the CRP comparisons: Pt.1 ($n = 7$, $p = 0.0009$), Pt.2 ($n = 4$, $p < 0.0001$), and Pt.3 ($n = 18$, $p = 0.028$), NOMID ($n = 11$, $p < 0.0001$) SAVI ($n = 7$, $p = $ ns), HC ($n = 5$) (left upper panel). For IL-6: Pt.1 ($n = 3$, $p = 0.0153$), Pt.2 ($n = 1$, $p = $ ns), Pt.3 ($n = 1$, $p = $ ns), NOMID ($n = 12$, $p < 0.0001$) and SAVI ($n = 6$, $p = 0.0019$), HC ($n = 26$) (left lower panel). For neutrophil and endothelial markers, p-values for respective comparisons with HC are: for sL-selectin: Pt.1 ($n = 2$, $p = 0.0186$), Pt.3 ($n = 1$, $p = $ ns), NOMID ($n = 12$, $p < 0.0001$) and SAVI ($n = 6$, $p = 0.0003$), HC ($n = 105$); for lipocalin: Pt.1 ($n = 2$, $p = 0.0169$), Pt.3 ($n = 1$, $p = $ ns), NOMID ($n = 12$, $p = $ ns) and SAVI ($n = 7$, $p = $ ns), HC ($n = 49$); for ICAM-1: Pt.1 ($n = 2$, $p = 0.0044$), Pt.3 ($n = 1$, $p = 0.0198$), NOMID ($n = 12$, $p = 0.0002$) and SAVI ($n = 7$, $p = 0.0001$), HC ($n = 26$); for sE-selectin: Pt.1 ($n = 2$, $p = 0.0260$), Pt.3 ($n = 1$, $p = $ ns), NOMID ($n = 12$, $p < 0.0001$) and SAVI ($n = 6$, $p = 0.0105$), HC ($n = 105$). Data are presented as median values ± interquartile ranges. *$p < 0.05$, **$p < 0.01$, ***$p < 0.001$, ****$p < 0.0001$ as determined by Kruskal–Wallis test. Exact significant p-values of patient vs. HC comparisons for each marker are stated above. Source data are provided as a Source data file.

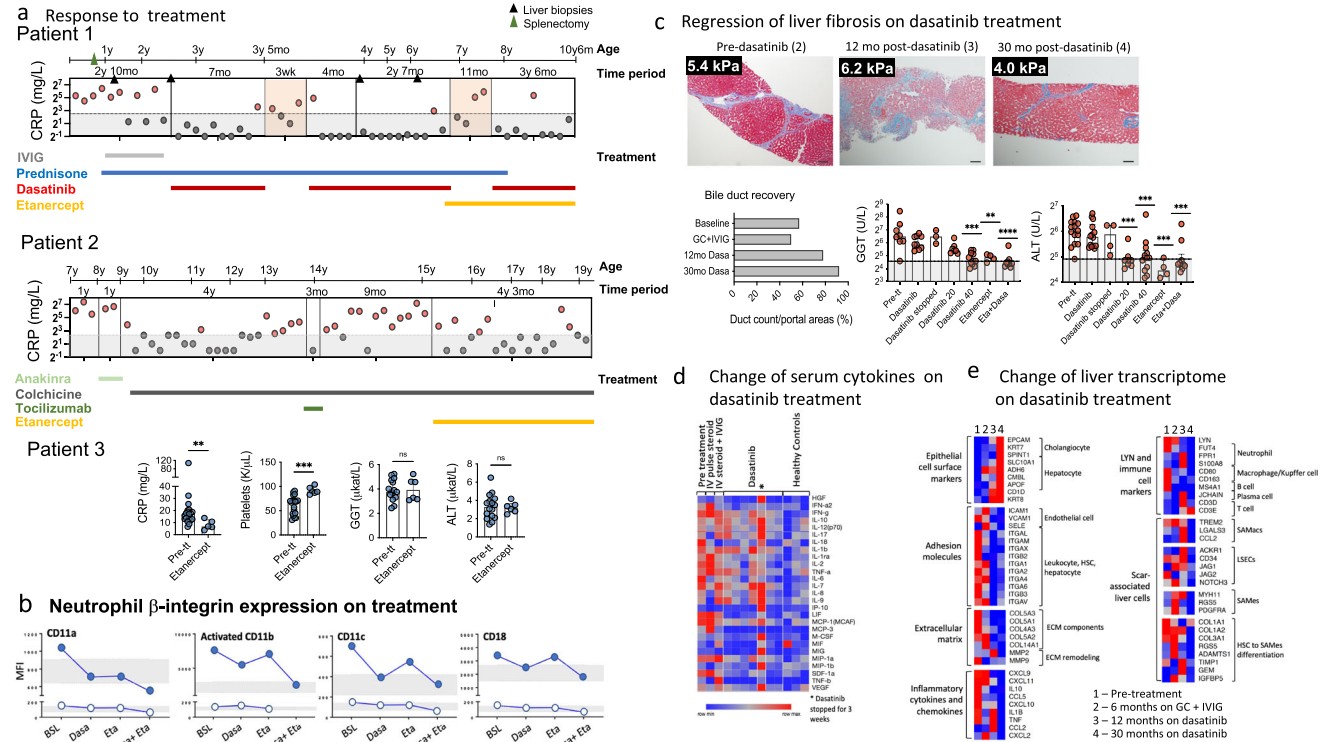

**Fig. 4 | Response to treatment in the three patients with gain-of-function mutations in _LYN_. a** Longitudinally obtained CRP levels are depicted for all patients stratified by treatment. C-reactive protein (CRP), liver function tests (LFTs), gamma-glutamyltransferase (GGT), and alanine aminotransferase (ALT). For Pt.3 comparisons between pre-treatment (pre-tt) and post etanercept treatment (etanercept) samples were depicted in scatter plots with bar graphs showing median and interquartile ranges. For CRP: pre-tt $n = 28$, etanercept $n = 5$, $p = 0.0056$; platelets, pre-tt $n = 28$, etanercept $n = 6$, $p = 0.0005$; GGT, pre-tt $n = 15$, etanercept $n = 6$, $p = 0.9568$; ALT, pre-tt $n = 16$, etanercept $n = 6$, $p = 0.9591$. **$p < 0.01$, ***$p < 0.001$ as determined by Mann–Whitney test with two-tailed _p_-value. ns, non-significant. **b** Pt.1 neutrophils were assessed at baseline and on dasatinib monotherapy or combination therapy with etanercept for expression of β2 integrins CD11a/LFA-1α, CD11b/Mac-1α, CD11c, and CD18/LFA-1β/Mac-1β. **c** Liver responses to various treatments for Pt.1 show improvement of bridging fibrosis in liver biopsies on dasatinib monotherapy (upper panels, scale bars, 100 μm), bile ductopenia (lower left panel) and the liver function markers, GGT and ALT (lower center and right panels). Scatter plot with bar graphs show median and interquartile ranges of pre-tt and variable post-treatment samples for GGT and ALT. For GGT and ALT, respectively: pre-tt ($n = 8$ and $n = 15$), dasatinib ($n = 9$ and $n = 14$), dasatinib stopped ($n = 3$ and $n = 4$), dasatinib 20 mg ($n = 7$ and $n = 8$, $p = 0.0005$), dasatinib 40 mg ($n = 11$, $p = 0.0002$ and $n = 13$, $p = 0.0004$), etanercept ($n = 4$, $p = 0.0074$ and $n = 4$, $p = 0.0002$), etanercept + dasatinib (eta+dasa) ($n = 9$, $p < 0.0001$ and $n = 10$, $p = 0.0002$). **$p < 0.01$, ***$p < 0.001$ as determined by Kruskal–Wallis test. **d** Heatmap depicts changes in serum cytokine and chemokine concentrations in longitudinally collected samples from Pt.1 compared to HC ($n = 3$). (*) sample obtained when dasatinib was temporarily stopped. **e** Heatmap of transcriptome analysis of 4 liver biopsies (Pt.1). A recovery of epithelial markers (e.g., cholangiocytes) and improvement of fibrosis and inflammation markers, of markers of liver sinusoidal endothelial cells (LSEC) activation, scar/fibrosis associated mesenchymal cells (SAMecs) and macrophages (SAMacs) on dasatinib treatment was observed. ECM extracellular matrix, HSC hepatic stellate cells, GC glucocorticosteroids, IVIG intravenous immunoglobulin. Source data are provided as a Source data file.

treatment with glucocorticosteroids and intravenous immunoglobulin (IVIG) (Patient 1), or IL-1 and IL-6 targeted treatment (Patient 2). In Patient 1, CRP levels were lower while on treatment with dasatinib monotherapy or dasatinib and etanercept combination therapy. Patient 2 had partial responses to colchicine, but recurrence of rash, fatigue, and systemic inflammation led to empiric treatment with the TNF inhibitor etanercept at the age of 14, which rapidly improved skin rashes, normalized acute phase reactants, and led to long-term remission (Fig. 4a). Patient 1 received glucocorticosteroids and IVIG at age 2 years and 2 months. After 6 months on treatment, he continued to experience painful rashes, and fatigue and a repeat liver biopsy showed progressive ductopenia involving 50% of portal areas and development of bridging fibrosis (Fig. 4c, Supplementary Fig. 8, Supplementary Table 3). The discovery of the gain-of-function (GOF) mutation in _LYN_ and the disease progression prompted treatment with the Src kinase inhibitor dasatinib, which blocks Lyn kinase, and gradually normalized CRP (Fig. 4a), β2 integrin expression on neutrophils (Fig. 4b), and LFTs (Fig. 4c), as well as inflammatory cytokines and chemokines in peripheral blood (Fig. 4d). Temporary discontinuation of dasatinib resulted in skin rashes with a rise in inflammatory markers

and gamma-glutamyltransferase (GGT), that resolved on reinstitution of dasatinib (Fig. 4a, c, d). After 30 months on dasatinib monotherapy, bile ducts were reconstituted, and liver fibrosis had regressed (Fig. 4c, Supplementary Fig. S8, Supplementary Table 3). A recovery of epithelial markers (e.g., cholangiocytes) and improvement of fibrosis and inflammation markers was also observed in liver transcriptome analyses of bulk RNA seq data from four liver biopsies from Patient 1 on dasatinib treatment Fig. 4e). ANA titers progressively decreased on dasatinib, and all autoantibodies initially detected turned negative either after the first course of corticosteroids or on dasatinib therapy (see Supplementary Results). The patient has been in inflammatory remission over the last 3.5 years. Patient 2 had CRP levels that were most consistently controlled on treatment with etanercept and colchicine. Patient 3 was started on etanercept at the age of 4 months and had partial improvement of systemic inflammation (CRP), thrombocytopenia, and direct hyperbilirubinemia but had persistently elevated LFTs (Fig. 4a).

Liver sinusoidal endothelial cells (LSEC) express Lyn kinase and are specialized, fenestrated endothelial cells that play a central role in liver homeostasis by regulating intrahepatic vascular tone, immune

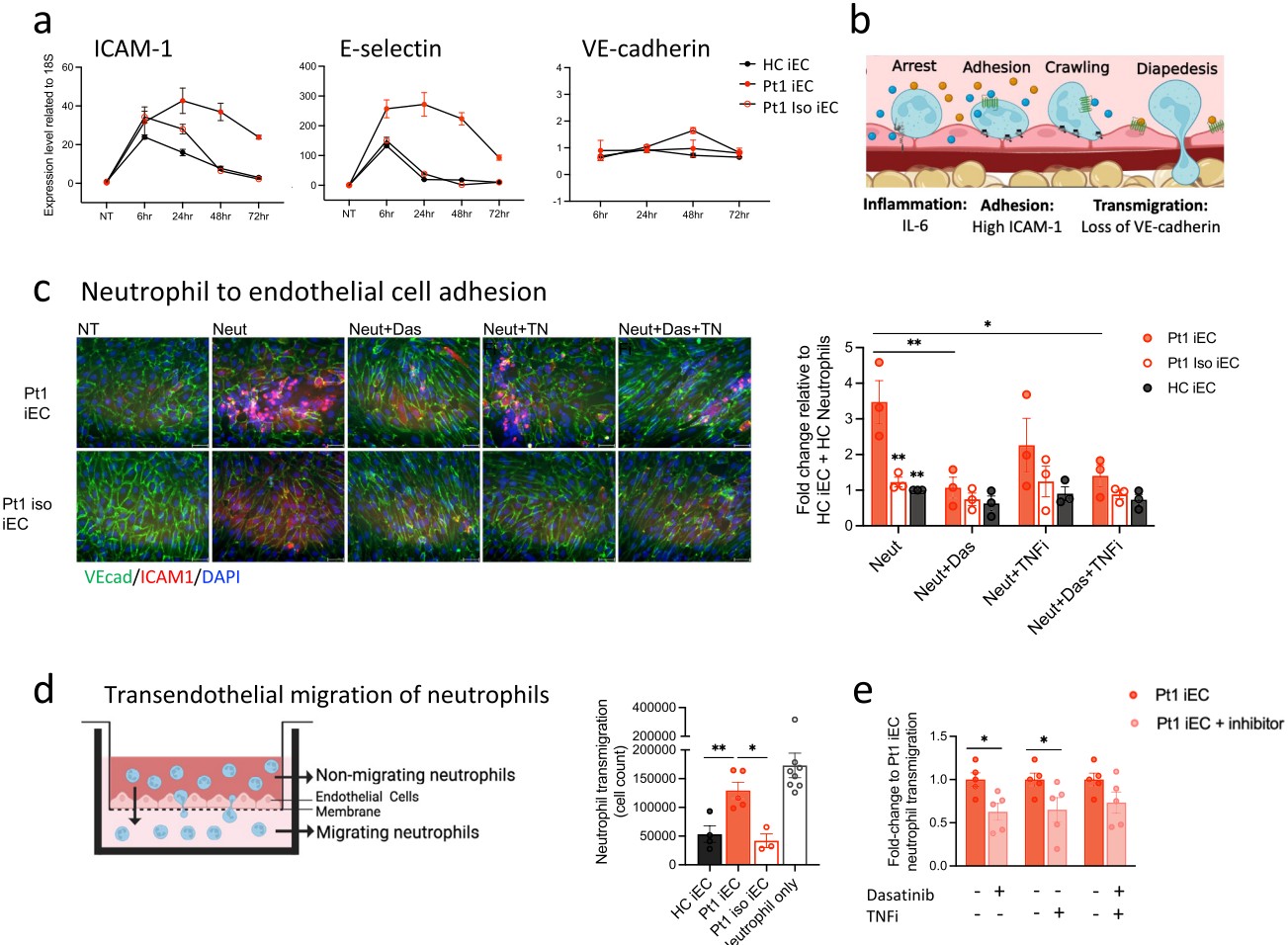

**Fig. 5 | Co-culture of neutrophils and induced endothelial cells (iECs) increases neutrophil adhesion, and transendothelial migration (TEM). a** mRNA expression of the endothelial cell markers ICAM-1, E-selectin, and VE-cadherin (VEcad) in Pt.1's iECs (Pt1 iECs), a genetically corrected isogenic iEC clone (Pt1 iso iEC) and a healthy control iEC (HC iEC) was quantified upon stimulation with IL-1β (10 ng/ml). The data were collected from *n* = 3 independently derived iEC cell lines for HC and Pt.1. One CRISPR/Cas9 edited Pt1 cell line was generated (iso iEC). Depicted are all biological and technical replicates for HC and Pt1 iEC, and technical replicates for Pt1 iso iEC. Data are presented as mean values ± SEM. **b** Schematic representation of neutrophil extravasation shows critical steps preceding diapedesis. *Created with BioRender.com.* **c** Co-cultures of HC neutrophils (Neut) with Pt.1 iECs (upper panels) or iso IECs (lower panels) for 48 h show neutrophil adhesion in mutant compared to iso iEC (*p* = 0.0058) and HC iEC (*p* = 0.0019) co-cultures, with the effect of treatment with dasatinib (Das, *p* = 0.0027), a TNF inhibitor (TNFi, *p* = 0.4089) or both (Das+TNFi, *p* = 0.0135) on neutrophil adhesion. Images are representative of 3

independent experiments. Scatter plot graph on y-axis indicates fold-change of the adhered neutrophil counts to HC iEC and HC neutrophil co-culture counts. Mean ± SEM of 3 technical replicates from 3 separate experiments are depicted. *\*p* < 0.05, *\*\*p* = 0.01, as determined by two-way ANOVA test. Scale bars, 50 μm. **d** Neutrophil transendothelial migration (TEM) is quantified after 24 h in an endothelial cell transwell migration assay. Data are presented as mean ± SEM from biological replicates of: HC iEC (*n* = 4), Pt1 iEC (*n* = 5), Pt1 iso iEC (*n* = 3), neutrophils only (*n* = 8). *\*p* = 0.0179, *\*\*p* = 0.0079, as determined by Mann–Whitney test with one-tailed *p*-value. *Left panel was created with BioRender.com.* **e** TEM of neutrophils is inhibited by dasatinib (*p* = 0.0158), or the TNF inhibitor (TNFi) (*p* = 0.0496); y-axis represents fold-change of the migrated neutrophil counts compared to Pt1 iEC and HC neutrophil co-culture counts. Mean ± SEM from 5 biological replicates from 3 separate experiments. *\*p* < 0.05, as determined by paired t-test with one-tailed *p*-value. Source data are provided as a Source data file.

cell function, and quiescence of hepatic stellate cells (HSCs)[18–21]. During sustained hepatic injury, LSECs regulate the development of fibrosis[22] through activation of HSC that differentiate into scar-producing myofibroblasts[23]. Fibrogenic LSEC and HSC signatures have been described in human and murine liver biopsies[23,24]. We analyzed these transcriptional signatures associated with liver fibrosis in the four liver biopsies from Patient 1 (Fig. 4e) and saw reduction in markers of LSEC activation including the fibrogenic LSEC markers *CD34*, and *ACKR1* and the NOTCH3-JAG1 axis that regulates HSCs activation as well as of scar/fibrosis associated mesenchymal cells (SAMec) and macrophages (SAMacs), which were upregulated in biopsies with bridging fibrosis and decreased on dasatinib treatment. Expression of genes encoding inflammatory cytokines, chemokines, adhesion molecules (CAMs and integrins), extracellular matrix genes,

and those that drive HSC differentiation into mesenchymal cells also decreased. In contrast, gene transcription of epithelial cell markers increased, thus corroborating remodeling during fibrinolysis and recovery from ductopenia (Fig. 4e).

### Effect of *LYN* GOF mutation on neutrophil endothelial cell interaction and neutrophil transendothelial migration (TEM)

The clinical phenotype and the upregulation of β2-integrin function on neutrophils, and of ICAM-1 on skin endothelial cells and in the liver biopsies combined with the treatment response to dasatinib suggested a pivotal role of Lyn GOF mutations in endothelial dysregulation and as driver of vasculitis and liver fibrosis. We generated induced pluripotent stem (iPS) cell-derived endothelial cells (iECs), and genetically corrected (isogenic) iECs from Patient 1, and iECs

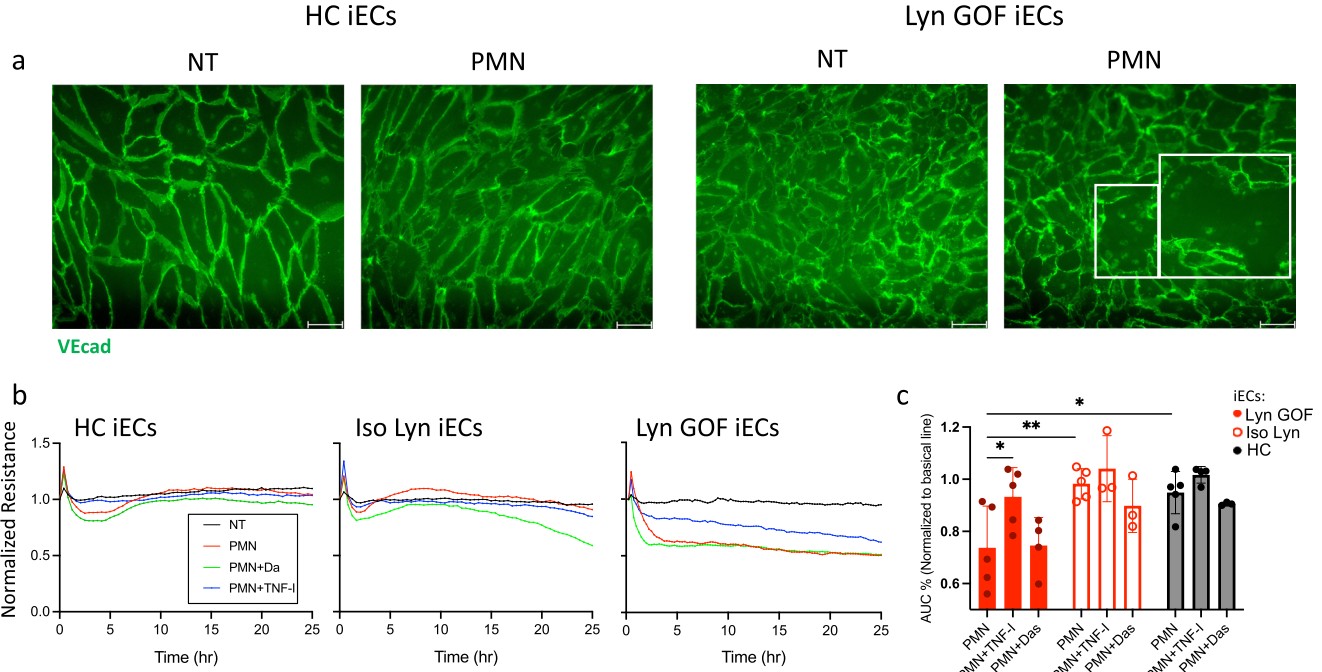

**Fig. 6 | Co-culture of neutrophils decreases transendothelial electrical resistance by electric cell-substrate impedance sensing (ECIS) in mutant but not isogenic iECs. a** Co-culture of neutrophils with iECs leads to decreased VE-cadherin staining (green) in Pt.1 iECs (Lyn GOF iEC, white boxes) compared to healthy control (HC) iECs. Scale bars, 50 µm. **b**, **c** ECIS data presentation of baseline absolute resistance (ohm, Ω) in iECs from Lyn GOF and isogenic control (iso Lyn) as well as HC iECs. Real-time ECIS measurement of transendothelial electrical resistance (TEER) across the endothelial monolayer upon co-culture with HC neutrophils (= polymorphonuclear cells, PMN) shows a decrease in Lyn GOF iECs compared to HC ($p = 0.0240$) and iso Lyn iEC ($p = 0.0099$). Treatment with a TNF inhibitor (TNF-I, $p = 0.0126$) improved TEER in Lyn GOF iEC. Absolute values were normalized to the

last reading before the additional treatment. Black line: iEC alone; red line: iEC and PMN added; blue line: PMN plus TNF-I; green line: PMN plus dasatinib. The changes and influence of resistance were analyzed as area under curve (AUC). Scatter plot with bar graph depicts mean values ± SEM of changes of resistance in iECs. The data were collected from $n = 3$ biologically independent samples for HC and Pt1 iEC and $n = 1$ biologically independent sample for Pt1 iso iEC. *$p < 0.05$, **$p < 0.01$ as determined by two-way ANOVA for comparison between Lyn GOF PMN and PMN + TNF-I, and ordinary one-way ANOVA for comparison between Lyn GOF PMN and HC or iso Lyn. Turkey multiple comparisons post-test was used in all analyses. Source data are provided as a Source data file.

from 3 healthy controls, and assessed the role of the Lyn kinase mutations on endothelial immune activation, adhesion, and endothelial transmigration, thus avoiding Lyn overexpression, which can activate downstream pathways[25] (Fig. 5a, b). Cytokine (IL-1β) stimulation of mutant iECs resulted in increased and prolonged mRNA expression of ICAM-1 and E-selectin but not of VE-cadherin compared to controls (Fig. 5a). Compared to isogenic patient derived but genetically corrected iEC controls, treatment of mutant iECs with dasatinib did not correct prolonged transcription in the mutant iECs, which was reversed in the isogenic controls; dasatinib treatment did, however, reduce ICAM-1 protein secretion (Supplementary Fig. 9a–c). Co-culture of patient but not healthy control neutrophils with mutant or isogenic iECs led to elevated IL-6 levels, which was similarly elevated in co-culture with mutant and isogenic iECs. IL-6 secretion was blocked by dasatinib and less by the TNF inhibitor (Supplementary Fig. 9d). To separate the effect of the Lyn kinase mutation on mutant endothelial cells and mutant neutrophils, the subsequent co-culture experiments were conducted with healthy control (HC) neutrophils. Co-culture of HC neutrophils with mutant but not isogenic or HC iECs led to clustered expression of ICAM-1 beneath areas of firmly adherent neutrophils (Fig. 5c). Neutrophil adhesion and ICAM-1 clusters were significantly blocked with dasatinib and less with the TNF inhibitor at 48 h (Fig. 5c). To evaluate neutrophil diapedesis, we assessed HC neutrophil transendothelial migration (TEM); TEM through patient iECs was significantly increased compared to isogenic or HC iECs (Fig. 5d) and improved with treatment with dasatinib or the TNF inhibitor (Fig. 5e).

## Effect of *LYN* GOF mutation on vascular permeability measured by electric cell-substrate impedance sensing (ECIS)

In areas of rich neutrophil adherence to mutant but not HC iEC monolayers, VE-cadherin expression was focally reduced (Fig. 6a and Supplementary Fig. 10), which suggested disruptions of cell–cell contacts and increased vascular permeability. The internalization of VE-cadherin into subcellular compartments and subsequent recycling or degradation and the concomitant disruptions of cell–cell contacts have also been linked to increased vascular permeability[26], which can be assessed using transendothelial electrical resistance (TEER)[27]. TEER was decreased in mutant iEC layers but not isogenic iECs or HC iECs upon co-culture with HC polymorphonuclear leukocytes (PMNs) that was significantly improved with the TNF inhibitor, but not with dasatinib. However, the genetic correction normalized the TEER, suggesting off target effects of dasatinib on endothelial cell permeability. Together, these data suggest complementary effects of dasatinib on neutrophil recruitment and adhesion to endothelial cells and TEM, and of TNF inhibition in improving endothelial barrier function, suggesting that both drugs may be effective in improving neutrophilic small vessel vasculitis[28].

## Discussion
In summary, we describe an autoinflammatory disease, which is caused by de novo GOF mutations in *LYN*, the gene that encodes Lyn kinase. We propose to call this syndrome Lyn kinase-associated vasculopathy and liver fibrosis (LAVLI). Disease-causing mutations eliminate an inhibitory tyrosine (Y) at position p.Y508 that results in constitutively active Lyn kinase[29] and a severe clinical phenotype of cutaneous small

vessel vasculitis and liver fibrosis. Functional data illustrate a pivotal role of active Lyn kinase in regulating endothelial activation, neutrophil adhesion and transendothelial migration, partly through upregulation of β2 integrins on neutrophils[30,31], and adhesion molecule expression (i.e., ICAM-1) on endothelial cells[32].

The clinical presentation of Lyn GOF mutations as autoinflammatory disease with sterile neutrophilic small vessel vasculitis was unexpected, as mice engineered to carry the p.Y508F mutation, that is disease-causing in Patient 2, develop high-titer autoantibodies and glomerulonephritis reminiscent of systemic lupus erythematosus[29]. Although clinical autoantibody tests have been negative in all 3 patients on treatment, Patient 1 had low titer autoantibodies at presentation and Patient 2 developed transiently elevated ANA autoantibodies in a research setting of ANA testing by Hep2 cell staining. Furthermore, B cell stimulation assays identified defects in central and peripheral B cell tolerance (Supplementary Results and Supplementary Figs. 11–13). Patients with GOF Lyn kinase mutations may thus be at greater risk for the development of autoimmunity in the future and should be closely monitored.

In contrast to Patients 1 and 3 with the truncating mutations, Patient 2 with the *LYN* missense mutation, p.Y508F, did not develop liver disease suggesting that the truncating mutations that eliminate the 5 terminal amino acids may confer more severe disease. The C-terminal tail interacts with the SH2 domain in all Src kinases, including in Lyn kinase[33], which restricts their activity. Phosphorylation-independent low-affinity interactions of the SH2 domains with the C-terminal domain have been observed in vitro[34] and could restrict Lyn kinase activity more in patients with the missense mutation thus resulting in a milder phenotype. A dose-dependent effect of Lyn kinase on liver fibrosis has been evaluated in a murine model of carbon tetrachloride (CCl4) induced liver fibrosis, that was reverted by treatment with a Src kinase inhibitor which attenuated HSC activation[25]. As hepatic stellate cells do not express Lyn kinase, their differentiation into scar producing fibroblasts is likely driven by inflammatory cytokines produced by neighboring cells including liver sinusoidal endothelial cells (LSECs) and Kupffer cells[35].

The association of elevated c-Src activity with cancer progression[36] and of mutations in c-Src in driving malignant progression of colon cancer[6] led to studies assessing the transforming potential of Lyn kinase. These studies suggest a decreased tumorigenic potential of Lyn compared to c-Src[33,37] and failed to link increased Lyn kinase activation in B cells to lymphoproliferative disease[38].

Instead, GOF mutations in *HCK* and *LYN* present with perinatal-onset systemic inflammation, neutrophilic small vessel (or leukocytoclastic) vasculitis, and lung inflammation with a *HCK* GOF mutation[10] and liver fibrosis with *LYN* GOF, respectively. A recent case report of a 3-year-old girl with a *LYN* missense mutation, p.Y508H, who presents with urticarial rash and systemic inflammation is largely consistent with the LAVLI phenotype described[39]. However, the findings of dysmorphic features and developmental delay that are described in that patient point to potential involvement of additional genetic factors.

Our clinical, functional, and the expression data suggest tissue specificity of the Src kinases, however, currently available inhibitors including dasatinib, target pan-Src kinases. In vitro, the treatment with the pan-Src kinase inhibitor dasatinib does not result in full correction of endothelial and neutrophil dysfunction caused by mutant Lyn, which is, however, achieved with isogenic gene corrected iECs (Fig. 6 and Supplementary Fig. 9)[40]. These discrepancies likely reflect the fact that dasatinib blocks a large number of kinases including other regulatory Src kinases that are known to collaborate in regulating endothelial homeostasis[12,41]. Supported by the organ-specific disease manifestations revealed by the monogenic defects that affect the function of the inhibitory tyrosine in the C-terminal tail of Src, Hck, and Lyn[3–5], it is intriguing to speculate that engineering inhibitors against specific Src family kinase members (i.e., Lyn kinase only) may provide unique targeted therapeutic strategies by increasing tissue specificity and reducing unwanted side effects.

Our study identifies Lyn kinase as potential treatment target[42] in small vessel vasculitis and early-onset liver fibrosis and illustrates the importance of a genetic diagnosis in patients with recalcitrant inflammatory disease. Whether Lyn kinase-specific inhibition would be a treatment strategy in a wider spectrum of non-syndromic forms of neutrophilic vasculitis will require further evaluations.

## Methods

### Study participants
All research investigations were done as part of protocol 17-I-0016/NCT02974595 and as such were approved by the NIAID and NIAMS/NIDDK Institutional Review Boards. The patients were referred to the NIH between 2013 and 2021. For Patient 3, the research investigations were also approved by the Institutional Review Board of the Second Faculty of Medicine, Charles University and University Hospital Motol. Written informed consent was obtained from the parents of all patients/healthy controls involved, and assent was obtained from the patients/healthy controls where indicated. The index patient was referred for unexplained systemic inflammation, and recurrent fever and rashes and underwent trio analysis at the NIH Clinical Center. The second patient was identified in Great Ormond Street Hospital for Children, London, UK and the third patient in the University Hospital Motol, Prague, Czech Republic. Clinical, laboratory and imaging data and unstained slides from clinically indicated skin, gastrointestinal and liver biopsies were obtained and stained for research markers. The data collection period for some parameters is from birth to April 2022 for all patients. The NIH protocol enrolls parents and healthy siblings as controls. The control experiments were conducted with blood from the patients' parents and or controls from the blood bank approved under the protocol. The authors affirm that the patients (or their parents/legal guardians) provided written informed consent for publication of the images in Fig. 2 and Supplementary Fig. 2 and the medical information included in this paper. Dasatinib was initially provided to Patient 1 under a compassionate use protocol. Etanercept was provided to treat the three patients' clinical phenotype on an off-label basis. A statement from all 3 patients or their parents regarding the effect of treatment are cited in the Supplementary Results section. The study was reported according to CARE guidelines and conducted in compliance with the Declaration of Helsinki principles.

### Genetic and functional analyses
The primary objectives of the natural history study, the study the patients were enrolled in, are to identify the genetic cause in patients with early-onset severe systemic inflammation, and to characterize the disease pathogenesis. We use trio WES or WGS and targeted genetics to identify and validate a genetic discovery and cellular models to mimic and interrogate immune dysregulation induced by a novel mutation in a clinically relevant tissue or cell model.

We performed genetic analyses including whole exome and Sanger sequencing of the patients and their parents. Genomic studies are detailed in the Supplementary Information file.

### Functional studies
#### Phosphorylation assay of lyn kinase and its substrates
**Cell culture.** HEK293FT cell line was purchased from Invitrogen (cat#R70007 Thermo Fisher Scientific, Waltham, MA, USA) and cultured in Dulbecco's modified Eagle's medium supplemented with 10% fetal bovine serum at 37 °C in 5% $CO_2$.

**Plasmid constructs.** The plasmid construct encoding wild-type *LYN* in pcDNA3 vector was kindly provided by S. Watson, University of Birmingham, Birmingham, United Kingdom. Plasmids expressing mutant *LYN* (*LYN*, c.1519 C > T; *LYN*, c.1523 A > T and *LYN*, c.1524 C > G) were

generated by site-directed mutagenesis (QuikChange™IIXL Site-Directed Mutagenesis kit, Agilent Technologies, Santa Clara, CA, USA) using the following primer pairs: 5′-ctgctgctggtattacc cttccgtggctg-3′ and 5′-cagccacggaagggtaataccagcagcag-3′ for c.1519 C > T; 5′-ctgctgctggaattgcccttccgtggctgt-3′ and 5′-acagccacggaagggc aattccagcagcag-3′ for c.1523 A > T; 5′-gctgctgctgctattgcccttccgtggct-3′ and 5′-agccacggaagggcaatagcagcagcagc-3′ for c.1524 C > G). Successful introduction of the mutations was confirmed by Sanger sequencing. Plasmid encoding SKAP2 with N-terminal FLAG tag was generated by PCR from HL-60 cell line cDNA with these primers: 5′-ctgaattctcc caacccccagcagcacctc-3′ and 5′-atggatcctcaaatatcatacatctccattatgtagg-3′, followed by cloning into EcoRI and BamHI sites of pFLAG-CMV-2 vector and Sanger sequencing.

**Transfection.** Plasmids encoding wild-type or mutant *LYN* or their combination with a plasmid encoding the known Lyn kinase substrate *SCIMP*[14] and *SKAP2* were co-transfected to the HEK293FT cell line using lipofectamine 2000 transfection reagent (Thermo Fisher Scientific, Waltham MA, USA) according to manufacturer's instructions.

**Cell lysis and immunoblotting.** 24 h after transfection, HEK293FT cells were lysed in SDS-PAGE sample buffer followed by 15 s sonication (amplitude 50, 4710 Ultrasonic Homogenizer, Cole Parmer) and immunoblotting with antibodies to phosphotyrosine (4G10 culture supernatant produced in house), phospho-Src Family Tyr416 (#2101, Cell Signaling Technology, Danvers, MA, USA), phospho-Lyn Tyr508 (#2731, Cell Signaling Technology, Danvers, MA, USA), rabbit polyclonal to Lyn (sc-015, Santa Cruz Biotechnology, Dallas, TX, USA), or monoclonal antibody to SCIMP (NVL-07, produced in house)[14].

**Immunoprecipitation and in vitro kinase assay.** To detect SKAP2 phosphorylation, HEK293FT cells transfected in 12-well plates with constructs encoding SKAP2 and *LYN* mutants as described above were lysed in 250 μl lysis buffer containing 1% n-Dodecyl-β-D-maltoside, 30 mM Tris-HCl pH 7.4, 120 mM NaCl, 2 mM KCl, 2 mM EDTA, 10% glycerol, cOmplete™ EDTA-free Protease Inhibitor Cocktail (Roche), and PhosSTOP™ phosphatase inhibitor cocktail (Roche). Lysates were cleared by centrifugation at 21,000 × *g* for 10 min at 4 °C. FLAG-tagged SKAP2 was immunoprecipitated with 5 μl anti-FLAG agarose beads (Merck, Sigma-Aldrich) for 2 h at 4 °C, followed by elution with 33 μl SDS-PAGE sample buffer. The immunoprecipitates were subjected to immunoblotting with anti-phosphotyrosine (4G10) and anti-FLAG M2 antibodies (Merck, Sigma-Aldrich, St. Louis, MO, USA), and the lysates to immunoblotting with antibodies against LYN (Santa Cruz Biotechnology, Dallas, TX, USA) and GAPDH (Merck, Sigma-Aldrich, St. Louis, MO, USA).

**Serum cytokine analysis.** Serum was collected from patients 1 and 3 and stored at −80 °C.

Patient 1: IL-6 and cytokines and chemokines depicted in Fig. 4c were measured in 10 serum samples from patient 1 and 3 healthy controls using the Bio-Plex ProTM Human Cytokine 27-Plex and 21-Plex Immunoassays (Bio-Rad, Hercules, CA, USA). Human cytokine standard group I and II were used for the standard curves. The sera were diluted 1:4 with the sample buffer and 50 μl of the diluted serum were used for the assays. All sera were analyzed simultaneously to avoid batch effects. Values below the limit of detection were reset at the limit of detection. Soluble ICAM-1, L-selectin, E-selectin, lipocalin/NGAL, MMP-9, lactoferrin, MPO, and RANTES were measured on customized, magnetic bead-based, multiplex assay (R&D Systems, Minneapolis, MN, USA) according to the manufacturers specifications for standards and dilutions. The magnetic beads were analyzed on Bio-Plex 3D instrumentation (Bio-Rad). Standard curves were analyzed using non-linear curve fitting and unknowns were calculated based on the derived equation. Samples that exceeded the highest standards were

reanalyzed more diluted until the values fell within the range of the known standards. Two control plasma samples and a control sample spiked with a known quantity of each analyte were analyzed on each plate to assess the inter-plate variation and to determine the effect of the biological matrix on the measurement of each analyte. For most analytes, the control samples had <20% variation from plate to plate, and the recoveries were generally >70%.

Patient 3: Serum levels of E-selectin, L-selectin, ICAM-1, and lipocalin were detected by ELISA (Abcam, Waltham, MA) in the samples of Patient 3 and 21 healthy controls. The absorbance was measured by ELx800UV absorbance microplate reader (Agilent Technologies). Serum IL-6 level was determined by routine in-house method (chemiluminescence), and the data were acquired using Immulite 2000 XPi (Siemens Healthcare, Erlangen, Germany).

Normalization procedure: In order to plot patients 1, 2, and 3's serum markers IL-6 (Patients 1, 2, and 3), ICAM-1, L-selectin, E-selectin and lipocalin (Patients 1 and 3) on the same graphs, we normalized data using the 2.5th and the 97.5th percentiles for the healthy controls (HC) for the respective assays that were used. Normalized values depicted in the graphs correspond to the analyte concentration minus 2.5%ile of HC, divided by the 97.5%ile minus 2.5%ile of HC, as follows:

$$\text{Normalized value} = \frac{\text{analyte conc.} - 2.5\%\text{ile of HC}}{97.5\%\text{ile} - 2.5\%\text{ile of HC}}$$

**Neutrophil characterization and functional assessment**
**PMN surface antigen expression.** Whole blood samples from healthy controls and patients were collected into EDTA tubes (Vacutainer; BD Biosciences). Aliquots (100 μl) of whole blood were incubated for 15 min at ambient (room) temperature with the following antibodies: CD11b, CD11b (activation epitope) (eBioscience), CD11a, CD11c, CD16, CD18, CD32 CD45, or CD63, CD62L, CD63, CD64 (BD Biosciences). The tubes were treated with OptiLyse C (Beckman Coulter Inc) for 10 min at room temperature to lyse the erythrocytes, washed twice, and then analyzed on the BD FACSCanto II cytometer (BD Biosciences). Expression, as measured by mean fluorescence intensity (MFI), was measured for each surface antigen.

**Detection of NETs in tissue.** NETs were detected in tissue as previously described[43]. Briefly, tissues were dehydrated with various ethanol dilutions. After antigen retrieval, samples were blocked and incubated with anti-citrullinated histone H4 (Millipore, 1:100) overnight. After a series of washes with PBS, tissues were incubated with Anti-rabbit Alexa Fluor 555 (Life Technologies) secondary antibody. Nuclei were counterstained with Hoechst for 10 min. Images were acquired on a Zeiss LSM780 confocal laser-scanner microscope.

**Generation of iECs and isogenic controls and functional studies.** The patient-specific induced pluripotent stem cells (iPSC) were generated from patient fibroblasts via activation of Yamanaka's transcription factors (Oct4, Sox2, Klf4, and C-Myc). By using CRISPR-Cas9 gene editing technology, we obtained the isogenic control iPSC line originating from patient-specific iPSC. We have also derived iPSC lines from healthy volunteers (control) who did not have the *LYN* mutation. The iPSC lines were differentiated into endothelial cells (iEC) following our previously published protocol[44]. In brief, iPSCs were seeded in Matrigel-coated plates at low-density supplement with TeSR-E8. After 24 h, culture cells were induced toward mesoderm progenitors in mesoderm differentiation medium for 6 days. Consequently, the CD31 positive cells from the induction culture were enriched by using CD31 magnetic beads, and further cultured and expanded in Collagen I coated plates or dishes with endothelial culture medium. For immunostaining, cells were fixed with 4% paraformaldehyde (PFA) and stained using antibodies for CD31, CD144, and von Willebrand factor (vWF). Nuclei were visualized with DAPI (Thermo Fisher Scientific).

Stained cells were photographed with a fluorescence microscopic system (Zeiss, Oberkochen, Germany).

A low-density lipoprotein (LDL) uptake assay was performed as described previously[44]. Briefly, the iECs were incubated with 10 μg/ml acetylated LDL (Ac-LDL) labeled with 1,1'-dioctadecyl-3,3,3',3'-tetra-methylindo-carbocyanine perchlorate (DiI-Ac-LDL) (Invitrogen, Catalog# L3484) for 4 h at 37 °C. The DiI-Ac-LDL uptake was assessed and quantified by fluorescent microscopy. Subsequently, cells were dissociated into single cells with TrypLE TM Express Enzyme. The data acquisition was performed on a MACSQuant Flow Cytometer (Miltenyi Biotec, Bergisch Gladbach, Germany) and the results were analyzed with FlowJo software (FlowJo, LLC).

### Co-culture assays

**Transmigration assay.** The transwell inserts of a 24-well plate were coated with fibronectin (31.25 μg/mL) for 30 min and followed seeding iPS-derived endothelial cells (iECs) at density of 60,000 cells/well in the upper insert. After iECs formed a confluent monolayer (24–48 h later), freshly isolated healthy control neutrophils at neutrophils/iECs 10:1 ratio were added to the top insert with 1:1 ratio mixed media (iEC media/RPMI with 10% FBS); the same media was added to the bottom chamber. The neutrophils that transmigrated to the bottom chamber were collected at 24 h after co-culture. Cells were counted with Bio-Rad cell counter.

**Neutrophil adhesion assay.** iECs in passage 3 were used for experiments. iECs were co-cultured with or without fresh isolated healthy control neutrophils at neutrophils/iECs 10:1 ratio for 48 h then fixed with 4% paraformaldehyde for 10 min at RT. The primary antibody VE-cadherin (1:100 Santa Cruz) and ICAM-1 (1:100, Abcam) were applied overnight at 4 °C, followed by the corresponding secondary antibody for 1 h at room temperature. DAPI (1:1000) was used for cell nuclei detection. Cells were imaged using a Zeiss inverted fluorescence microscope. Counting of adhered neutrophil nuclei via DAPI staining for adhesion was conducted on 10 random fields of view of 6 (Pt1 iso iEC & HC iEC) to 9 (Pt1 iEC) technical replicates conducted in 4 separate experiments. IL-6 in the co-culture supernatant was analyzed by ELISA (R&D Systems, Minneapolis, MN).

**Electric impedance spectroscopy (EIS) in assessing transendothelial electrical resistance (TEER).** The barrier function of endothelial cells was measured by using Electric Cell-substrate Impedance Sensing (ECIS) technology (Applied BioPhysics, Troy, NY, USA). ECIS Zθ technology is used to assess and quantify endothelial permeability. The ECIS arrays (8W10E+) were pretreated with 10 mM L-cysteine, followed by coating with fibronectin (Sigma). The iEC were seeded at 120,000 cells per well in 400 μL of iEC growth medium. The barrier integrity of iEC was monitored for 48 h when the cells were treated with dasatinib (10 nM, and 50 nM, Selleckchem, Houston, TX), TNFα inhibitor adalimumab (12 μg/ml), and dasatinib plus TNFα inhibitor. The multiple frequency data were collected for barrier modeling as recommended by Applied Biophysics. ECIS measurements were acquired from three independent biological repeats with three replicates per experiment. The panel shown in Fig. 6 is from one independent experiment with three replicates, which is representative of the three biological repeats. The results of all three independent biological repeats were graphed in panel c of the figure. Graphs were generated using GraphPad Prism software (GraphPad Software).

**Gene expression by RT-qPCR.** The iEC were seeded and cultured into collagen I coated 12-well plates at the density of 100,000 cells per well for overnight. Followed by treatment with IL-1β, total RNA of iEC was isolated by using RNeasy Mini Kits (Qiagen, Hilden, Germany). cDNA was synthesized by reverse transcription (RT) using Super Script™ III (Invitrogen, Waltham, MA, USA). RT-qPCR was performed using SYBR Green Premix on a Real-Time PCR Detection System (Bio-Rad). Assays were run in duplicate, and results were normalized to 18S ribosomal RNA expression. Primers used for RT-qPCR for ICAM-1 (*ICAM1*), E-selectin (*SELE*), and VE-Cadherin (*CDH5*) are shown in Supplementary Table 5.

**Tissue immunofluorescence staining.** Formalin-fixed, paraffin-embedded skin samples were deparaffinized with two 30 min washes in xylene, then a series of two 100% ethanol washes for 1 min each, and 1 min washes in 90%, 80%, 70%, 50% ethanol then water. Antigen retrieval was performed in a solution of 10 mM sodium citrate (pH 6.0) or 1 mM EDTA buffer (pH 8.0) for 40 min at 95 °C. Once cooled to room temperature (RT) samples were blocked for 1 h in 5% BSA, 20% donkey serum, and 0.1% Triton-X100 in PBS. Primary antibodies for CD31 (1:50, DAKO, Glostrup, Denmark), E-selectin (1:25, Abcam), ICAM-1 (1:50, Abcam), and vWF (1:200, DAKO) were applied overnight at 4 °C, followed by the corresponding secondary antibody for 1 h at room temperature. Slides were mounted using DAPI-containing mounting media (Vector Laboratories). Slides were imaged using a Zeiss LSM 510 META confocal microscope.

### Reporting summary

Further information on research design is available in the Nature Portfolio Reporting Summary linked to this article.

## Data availability

Next-generation sequencing data from patients 1, 2, and 3 are not publicly available due to patient privacy and confidentiality. Requests for data use for medical research should be directed to the corresponding author. Upon request, we will provide the data within two weeks provided that the requesting investigator will use appropriate protection of the patient's privacy according to data use agreement. The *LYN* (NM_002350.4) variants were deposited in ClinVar (https://www.ncbi.nlm.nih.gov/clinvar/) under accession numbers SCV002822944 for c.1519 C > T, p.Q507*; SCV002822945 for c.1523 A > T, p.Y508F; and SCV002822946 for c.1524 C > G, p.Y508*. The variants were also added to the Infevers database (https://infevers.umai-montpellier.fr/web/), The Registry for Autoinflammatory Disorders Mutations, shortly after publication. Source data are provided with this paper.

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

## Acknowledgements

The authors would like to thank Rachel Vantries, B.A., Gina Montealegre, M.D., Dawn Chapelle, C.R.N.P., Yan Huang, M.D., April Brundidge, R.N., Alexandra Liptakova, M.D., Anna Sediva, M.D., Ph.D., Milan Macek, Jr, M.D. Ph.D., and Zdenek Sumnik, M.D., Ph.D. for logistical help with patients and sample management; Ondrej Hrusak, M.D., Ph.D., Tomas Kalina, M.D. Ph.D., Milan Macek Jr., M.D., Ph.D., Zdenek Sumnik, M.D., Ph.D., Anna Sediva, M.D., Ph.D., Kveta Blahova, M.D. Ph.D., Alexandra Liptakova, M.D., Veronika Koukolska, M.D., Quan Yu, M.D, M.S., Karen Lau, M.S., for their contribution to this study, and the families for their cooperation. Funding was provided by the Intramural Research Program of the NIH, NIAID, NHLBI, NIAMS and the Clinical Center, the Ministry of Health of the Czech Republic (NV19-05-00332, NU20-05-00320, NV18-05-00162 and project for the conceptual development of research organization - University Hospital Motol 00064203), Ministry of Education, Youth and Sports of the Czech Republic (LM2018132 - to M.B. and P.P.), Charles University (PRIMUS/19/MED/04), and Programme EXCELES, No. LX22NPO5102.

## Author contributions

M.B., K.S.K., and R.G.M. designed the study. A.A.J., G.C., D.Y., T.B., D.B., F.B., A.R., L.K., B.M., I.D., E.O., P.P., C.C.-R., D.F., C.M., E.M., H.K., M.J.K., A.B., and D.K. performed the experiments. C.-C.R.L., D.E.K., K.R.C., S.M.H., S.P. provided the immunohistochemistry images. N.R., D.C., H.M., H.F., N.M., K.S., M.P., S.A., K.U., C.M.H., N.S., M.B., R.K., Z.P., L.P.,

D.K., V.C., L.S., and P.B. provided patient care, consultation, and clinical data collection. M.M. was responsible for the dasatinib compassionate use study. S.R.B, R.L.H., Z.D., A.H., S.M., and S.D.R. performed data analysis. R.G.M. and A.A.J. wrote the first draft of the manuscript and all authors revised and agreed with the manuscript content.

## Funding

## Competing interests
The authors declare no competing interests.

## Additional information

**Adriana A. de Jesus** [1,18], **Guibin Chen** [2,18], **Dan Yang** [2,18], **Tomas Brdicka** [3], **Natasha M. Ruth** [4], **David Bennin** [5], **Dita Cebecauerova** [6], **Hana Malcova** [6], **Helen Freeman** [7], **Neil Martin** [8], **Karel Svojgr** [6], **Murray H. Passo** [4], **Farzana Bhuyan** [1], **Sara Alehashemi** [1], **Andre T. Rastegar** [1], **Katsiaryna Uss** [1], **Lela Kardava** [9], **Bernadette Marrero** [1], **Iris Duric** [3], **Ebun Omoyinmi** [10], **Petra Peldova** [6], **Chyi-Chia Richard Lee** [11], **David E. Kleiner** [11], **Colleen M. Hadigan** [12], **Stephen M. Hewitt** [11], **Stefania Pittaluga** [11], **Carmelo Carmona-Rivera** [13], **Katherine R. Calvo** [12], **Nirali Shah** [11], **Miroslava Balascakova** [6], **Danielle L. Fink** [14], **Radana Kotalova** [6], **Zuzana Parackova** [6], **Lucie Peterkova** [6], **Daniela Kuzilkova** [6], **Vit Campr** [6], **Lucie Sramkova** [6], **Angelique Biancotto** [15], **Stephen R. Brooks** [13], **Cameron Manes** [16], **Eric Meffre** [16], **Rebecca L. Harper** [2], **Hyesun Kuehn** [12], **Mariana J. Kaplan** [13], **Paul Brogan** [10], **Sergio D. Rosenzweig** [12], **Melinda Merchant** [17], **Zuoming Deng** [13], **Anna Huttenlocher** [5], **Susan L. Moir** [9], **Douglas B. Kuhns** [14], **Manfred Boehm** [2,18], **Karolina Skvarova Kramarzova** [6,18] & **Raphaela Goldbach-Mansky** [1,18] ✉

[1]Translational Autoinflammatory Diseases Section (TADS), Laboratory of Clinical Immunology and Microbiology (LCIM), National Institute of Allergy and Infectious Diseases, National Institutes of Health, Bethesda, MD, USA. [2]National Heart, Lung, and Blood Institute, National Institutes of Health, Bethesda, MD, USA. [3]Laboratory of Leukocyte Signaling, Institute of Molecular Genetics of the Czech Academy of Sciences, Prague, Czech Republic. [4]Medical University of South Carolina, Charleston, SC, USA. [5]Departments of Pediatrics and Medical Microbiology and Immunology, University of Wisconsin-Madison, Madison, WI, USA. [6]Second Faculty of Medicine, Charles University/University Hospital Motol, Prague, Czech Republic. [7]Raigmore Hospital, Inverness, Scotland. [8]Royal Hospital for Children, Glasgow, Scotland. [9]Laboratory of Immunoregulation, National Institute of Allergy and Infectious Diseases, National Institutes of Health, Bethesda, MD, USA. [10]Great Ormond Street Hospital for Children NHS Foundation Trust, London, UK. [11]National Cancer Institute, National Institutes of Health, Bethesda, MD, USA. [12]Clinical Center, National Institutes of Health, Bethesda, MD, USA. [13]National Institute of Arthritis and Musculoskeletal and Skin Diseases, National Institutes of Health, Bethesda, MD, USA. [14]Collaborative Clinical Research Branch/Neutrophil Monitoring Laboratory, National Institute of Allergy and Infectious Diseases, National Institutes of Health, Frederick, MD, USA. [15]Sanofi Oncology, Vitry, France. [16]Yale University, New Haven, CT, USA. [17]AstraZeneca Research Based Biopharmaceutical Company, Waltham, MA, USA. [18]These authors contributed equally: Adriana A. de Jesus, Guibin Chen, Dan Yang, Manfred Boehm, Karolina Skvarova Kramarzova, Raphaela Goldbach-Mansky. ✉e-mail: goldbacr@mail.nih.gov

