## [Peer Review File · Nature Communications]

Constitutively active Lyn kinase causes a cutaneous small vessel vasculitis and liver fibrosis syndromeREVIEWER COMMENTS

Reviewer #1 (Remarks to the Author):

The manuscript "Constitutively active Lyn Kinase causes a cutaneous small vessel vasculitis and liver fibrosis" delineates a new autoinflammatory syndrome caused by gain of function mutations in Lyn Kinase. The authors identified three families with de novo mutations in LYN kinase all clustered within the same region, all with overlapping features including fevers, skin and liver inflammation. The authors go on to characterize in depth the clinical features of these patients and demonstrate dense neutrophilic infiltrates in the skin. They further characterize the mutations in vitro and demonstrate evidence that all three alleles lead to constitutive activation of the kinase and that patient cells have increased inflammatory cytokines. Most impressive of all is their work studying the treatment response of patients and tissue to inhibition of src kinase inhibition, further proving the importance of this pathway in disease and demonstrating an avenue for better control of patients. Finally, the authors use iPS models to better understand the role of endothelial cells in this newly defined inflammatory disease.

The paper is well written, experimentally sound, and will be of interest to the general audience of the journal. Most impressive of all is the authors ability to go from genetics, to molecular biology back to treating patients in one manuscript. However, there are some questions the authors should address prior to publication.

Major

- Can the authors comments on why the mutations all cluster in one region as compared to constitutive activation in cancer? Are there distinct mechanisms of activation unique to inflammation and cancer for this pathway? Do patients with LYN mutations in breast cancer have inflammatory disease? There should be a section added to the discussion on this point.
- For Figure 2B. SAM68 phosphorylation looks very non-specific and should be removed or optimized to remove non-specific bands. Why was this experiment performed in a different cell line? Why are the panels all different? I think the authors should demonstrate with recombinant protein the impact of these mutations if they haven't been shown in prior studies.
- Figure 5 should likely be removed from the paper since it is not well controlled or definitive data. The authors should include a separate patient's iPS cells and/or an edited WT cell line to better prove this effect. This data could be included as a supplement or removed if no additional controls are included, especially concerning is the modest effect in Figure 5C. If Figure 5C correction does not stop inflammation, what does the explanation mean? Either the system is not working properly or LYN kinase is not the cause? I think LYN is the cause, and I think the authors do too, so this figure should be carefully reassessed.

Minor

- Can the authors quantify the effect in Figure S4? There is a clear difference in the flow cytometry profile but it would be imp
- Can the authors spell out the LAVLI acronym? It is not listed in the discussion, just introduced? What does it stand for?

Reviewer #2 (Remarks to the Author):

The manuscript from deJesus et al is well written, obviously clinically important, and novel with respect to the discovery of a Lyn kinase gain of function mutation that results in vascular inflammation and liver fibrosis. I have mainly clarifying comments and suggestions to increase the relevance.

First and foremost, mutation of the inhibitory tyrosine in Src family kinases (SFKs), primarily pp60-cSrc, has been shown to create a constitutively active kinase. Phosphorylation of Y527/529 in Src by Csk (C-terminal Src kinase) is an established mechanism for negatively regulating SFKs. In absence

of Csk, or in its membrane associated scaffolding patterns including Cbp (Csk binding protein) and caveolin-1, SFK activity increases and resultant pathologies ensue (cancer of the colon, inflammation, and proliferation of various other cells, capillary hyperpermeability, etc). Of course the same goes for Src Y529F mutant that can not be phosphorylated. Thus, the paper describes this very same observation for Lyn, but does not relate this work to that established in the Src field.

Secondly, the substrates that constitutively-active (CA) Lyn phosphorylate and drive the phenotype are interrogated rather cursorily. Key substrates include ICAM-1 (which increases ICAM-1 expression and likely cleavage, thus increase in sICAM-1), FAK, paxillin (which increase motility), and caveolin-1 (which promotes Cav1 degradation and EMT/EndoMT), at least for CA-Src or persistent Src activation associated with inflammation. Persistent phosphorylation of some of these is associated with not only EMT/EndoMT, but fibrosis, which is clear from the RNAseq data showing increased expression of collagens, integrins, Notch3, etc). So what is the mechanism by which CA-Lyn promotes fibrosis in the liver? It may be indirect through activation of macrophage cytokine production, or direct effects (EndoMT or EMT) of LSECs and HSCs. This could be teased out further.

Reviewer #3 (Remarks to the Author):

The authors describe a novel autoinflammatory syndrome, caused by de novo GOF mutations in LYN, encoding Lyn kinase. They propose to call the syndrome Lyn kinase associated vasculopathy and liver fibrosis (LAVLI). The clinical presentation of this disease includes sterile neutrophilic small vessel vasculitis and systemic inflammation, but some patients also develop range of autoantibodies.

Whilst this manuscript was under review, another group based in Paris, France, has published an article describing a patient with the same clinical syndrome (De novo gain-of-function variations in LYN lead to an early onset systemic autoinflammatory disorder) doi: 10.1002/art.42354. This publication also included some genetic and clinical details of patients described in the present manuscript. These patients have been presented previously at scientific meetings but limited findings, until now, have been published in abstract format only. Publication of the article by the French group will certainly affect the novelty of the findings presented here. This manuscript will need to be edited to include the newest publication and include a discussion of their findings.

Nevertheless, the authors of the present manuscript offer much more solid evidence for the pathological effects of the novel LYN variants and provide some new biological insights worthwhile reporting.

However, there are some aspects which do require revision.

The authors seem to have largely neglected the autoimmune features that some of these patients have. The potential pathogenic role of B cells in the disease pathogenesis has also been minimised. The clinical features are suggestive of a condition that has overlapping autoinflammatory >>autoimmune characteristics. It is difficult to completely ignore presence of autoantibodies, even if this is transient. Unlike autoinflammatory diseases, autoimmunity might develop fully years after autoantibodies are first detected. It is uncertain if these patients will go on to develop autoimmune complication later in adulthood, particularly given the role of Lyn in B cell signalling and PI3K pathway activation in these cells. In my opinion, this condition might turn out to have many similarities with HA20, considering the cell-specific effects of Lyn, which will drive disease manifestations, some of which are probably still to be discovered. This needs to be addressed in discussion at least if no additional functional studies are possible. Therefore, although the acronym LAVLI sounds catchy, in the opinion of this reviewer is that it is not entirely appropriate in this case.

REVIEWER COMMENTS

Reviewer #1 (Remarks to the Author):

The manuscript “Constitutively active Lyn Kinase causes a cutaneous small vessel vasculitis and liver fibrosis” delineates a new autoinflammatory syndrome caused by gain of function mutations in Lyn Kinase. The authors identified three families with de novo mutations in LYN kinase all clustered within the same region, all with overlapping features including fevers, skin and liver inflammation. The authors go on to characterize in depth the clinical features of these patients and demonstrate dense neutrophilic infiltrates in the skin. They further characterize the mutations in vitro and demonstrate evidence that all three alleles lead to constitutive activation of the kinase and that patient cells have increased inflammatory cytokines. Most impressive of all is their work studying the treatment response of patients and tissue to inhibition of src kinase inhibition, further proving the importance of this pathway in disease and demonstrating an avenue for better control of patients. Finally, the authors use iPS models to better understand the role of endothelial cells in this newly defined inflammatory disease.

The paper is well written, experimentally sound, and will be of interest to the general audience of the journal. Most impressive of all is the authors ability to go from genetics, to molecular biology back to treating patients in one manuscript. However, there are some questions the authors should address prior to publication.

Major:

Comment 1:

Can the authors comments on why the mutations all cluster in one region as compared to constitutive activation in cancer? Are there distinct mechanisms of activation unique to inflammation and cancer for this pathway? Do patients with LYN mutations in breast cancer have inflammatory disease? There should be a section added to the discussion on this point.

Response:

We would like to thank the reviewer for the comments and would like to clarify that the mutations we see in the children are germline mutations. In contrast, most mutations in cancer are somatic in the transformed cells only and are often present in the context of other somatic mutations or cancer-causing germline mutations. Thus, the impact of activating Src kinase mutations in these cancer patients is restricted to the tumor cells that harbor the mutations. In the patients with the Lyn kinase gain-of-function (GOF) mutations, the mutation is germline and allows to assess the organ specific disease manifestations associated with a germline mutation. **We have clarified this issue and included and discussed published experimental data that reported on the potential of various Src kinase family members to trigger malignant transformation; these findings are added in the discussion section. There are reports that the malignant transformation potential for Lyn kinase is weaker than that for Src kinase. Another study finds that Lyn kinase activation was less associated with the development of lymphoproliferative disease and lymphoma/leukemia (Cai H et al. *Proc Natl Acad Sci U S A*.**

2011; Kohlhas V et al. *Blood Adv.* 2020). The clarification of this subject is in the discussion section starting with: " The discovery of *SRC* revealed the first human proto-oncogene,....."

Comment 2:

For Figure 2B. SAM68 phosphorylation looks very non-specific and should be removed or optimized to remove non-specific bands. Why was this experiment performed in a different cell line? Why are the panels all different? I think the authors should demonstrate with recombinant protein the impact of these mutations if they haven't been shown in prior studies.

Response:

The original Western blots in PLB-985 cells transfected with mutant and wildtype *LYN* were performed when we only knew of 2 patients. They were designed to show that there is an increase in phosphorylation of downstream targets of Lyn kinase. Our results indeed show an increased phosphorylation pattern indicating that there are many downstream targets that are hyperphosphorylated including SAM68 which is indicated by the arrow. The later experiments included all 3 mutations. **To avoid confusion, we moved the SAM68 phosphorylation figure panel to the supplement.** To clarify, experiments performed before inclusion (and birth) of the 3rd patient were done in PLB-985 and not HEK293FT cells. With the inclusion of the last patient, we assessed phosphorylation of Lyn kinase itself and of the two downstream targets Scimp and Skap2.

Comment 3:

Figure 5 should likely be removed from the paper since it is not well controlled or definitive data. The authors should include a separate patient's iPS cells and/or an edited WT cell line to better prove this effect. This data could be included as a supplement or removed if no additional controls are included, especially concerning is the modest effect in Figure 5C. If Figure 5C correction does not stop inflammation, what does the explanation mean? Either the system is not working properly or *LYN* kinase is not the cause? I think *LYN* is the cause, and I think the authors do too, so this figure should be carefully reassessed.

Response:

We apologize for not clarifying in the legend of figure 5C that the incubation in this experiment only is with patient neutrophils or with healthy control neutrophils; all other experiments were conducted with healthy control neutrophils to separate the effect of mutant endothelial cells from the effect of mutant neutrophils. The experiment shows a significant contribution of activated (patient) neutrophils to the inflammatory response. The cytokine release of IL-6 is equally potent when patient neutrophils were incubated with isogenic iECs or with Patient 1 derived mutant iECs. These experiments are challenging as they can only be performed when the patient is seen at the NIH as neutrophils cannot be frozen. To separate the effect of the mutation in iECs we conducted all other experiments with healthy control neutrophils. **We have removed the figure panel from the main figure and placed it in the supplement** (Supplementary Figure 9d) where we expanded on the figure legend to clarify that the goal of the experiment was to show the impact of activated mutant neutrophils on endothelial cells. We show similar IL-6 cytokine elevation when patient neutrophils are cultured with wildtype or

mutant iECs, which is absent when the iECs were cultured with healthy control neutrophils. We observed significant clumping of patient neutrophils, which prevented evaluation of the transmigration of mutant neutrophils. Our co-culture experiments with healthy control neutrophils show the significant impact the mutation has on endothelial cell function and demonstrate the variable effect of TNF inhibitor and dasatinib treatment on endothelial ICAM-1 capping and transmigration in vitro.

Minor

Comment 4:

Can the authors quantify the effect in Figure S4? There is a clear difference in the flow cytometry profile but it would be imp

Response:

We have added the quantification of the inhibitory effect of the respective inhibitors dasatinib or PP2 in Supplementary Figure S5 and updated the figure legend to describe the method used: “.....To quantify the effect of the inhibitor, we used fold induction of treated over untreated (DMSO alone) sample. The % decrease or increase is relative to absence of inhibitor or relative to healthy control”.

Comment 5:

Can the authors spell out the LAVLI acronym? It is not listed in the discussion, just introduced? What does it stand for?

Response:

LAVLI stands for LYN associated vasculitis and liver fibrosis. **We have introduced the proposed new disease name in the first paragraph of the discussion.**

Reviewer #2 (Remarks to the Author):

The manuscript from deJesus et al is well written, obviously clinically important, and novel with respect to the discovery of a Lyn kinase gain of function mutation that results in vascular inflammation and liver fibrosis. I have mainly clarifying comments and suggestions to increase the relevance.

Comment 1:

First and foremost, mutation of the inhibitory tyrosine in Src family kinases (SFKs), primarily pp60-cSrc, has been shown to create a constitutively active kinase. Phosphorylation of Y527/529 in Src by Csk (C-terminal Src kinase) is an established mechanism for negatively regulating SFKs. In absence of Csk, or in its membrane associated scaffolding patterns including Cbp (Csk binding protein) and caveolin-1, SFK activity increases and resultant pathologies ensue (cancer of the colon, inflammation, and proliferation of various other cells, capillary hyperpermeability, etc). Of course the same goes for Src Y529F mutant that cannot be

phosphorylated. Thus, the paper describes this very same observation for Lyn, but does not relate this work to that established in the Src field.

Response:

We thank the reviewer for the comments. We have added a figure in the manuscript that depicts the mechanism that regulates all Src kinase family members, including Lyn kinase, by Csk (Okada M et al. *Int J Biol Sci.* 2012) as explanatory supplementary figure (Supplementary Figure 3a) that illustrates the ubiquitous Src family kinase regulation by Csk and Cpb as outlined by the reviewer. We have discussed the differential effect of the various Src kinase family members in the discussion section. We also provide data and the suggestion that specific inhibition of Lyn kinase may allow to specifically target the Lyn kinase activity only and thus reduce off target side effects of a pan-Src kinase inhibitor.

Comment 2:

Secondly, the substrates that constitutively-active (CA) Lyn phosphorylate and drive the phenotype are interrogated rather cursorily. Key substrates include ICAM-1 (which increases ICAM-1 expression and likely cleavage, thus increase in sICAM-1), FAK, paxillin (which increase motility), and caveolin-1 (which promotes Cav1 degradation and EMT/EndoMT), at least for CA-Src or persistent Src activation associated with inflammation. Persistent phosphorylation of some of these is associated with not only EMT/EndoMT, but fibrosis, which is clear from the RNAseq data showing increased expression of collagens, integrins, Notch3, etc). So what is the mechanism by which CA-Lyn promotes fibrosis in the liver ? It may be indirect through activation of macrophage cytokine production, or direct effects (EndoMT or EMT) of LSECs and HSCs. This could be teased out further.

Response:

We thank the reviewer for these remarks. The exploration of the mechanism of liver disease cannot be addressed experimentally, as we do not have an *ex vivo* model of liver disease. However, we have clarified that Lyn kinase is not expressed in hepatic stellate cells (HSCs) but Lyn kinase is expressed in endothelial and/or Kupffer cells (on histopathology). The expression pattern and the transcriptional data are suggestive of an indirect effect on HSCs differentiation by activated endothelial and/or Kupffer cells that generate a pro-fibrotic environment by secretion of inflammatory cytokines and chemokines and other profibrotic mediators. The transcriptional profile in the serial liver biopsies is consistent with such a model; transcriptional upregulation of markers of inflammation including selected cytokines and chemokines is normalized with treatment. Furthermore, fibrogenic LSEC markers (i.e., *CD34*, and *ACKR1* and the NOTCH3-JAG1 axis) and markers of HSC differentiation into mesenchymal cells (i.e., *COL1A1*, *COL1A2* and *COL3A1*) are present at the same time in the liver biopsy samples and decrease with treatment. These findings are consistent with an indirect effect on HSC and liver fibrosis. We clarified these points in the discussion in the section starting with: **“...As hepatic stellate cells do not express Lyn kinase, their differentiation into scar producing fibroblasts is likely driven by inflammatory cytokines produced by neighboring cells including liver sinusoidal endothelial cells (LSECs) and Kupffer cells (Higashi T et al. *Adv Drug Deliv Rev.* 2017).”**

Reviewer #3 (Remarks to the Author):

The authors describe a novel autoinflammatory syndrome, caused by de novo GOF mutations in LYN, encoding Lyn kinase. They propose to call the syndrome Lyn kinase associated vasculopathy and liver fibrosis (LAVLI). The clinical presentation of this disease includes sterile neutrophilic small vessel vasculitis and systemic inflammation, but some patients also develop range of autoantibodies.

Comment 1:

Whilst this manuscript was under review, another group based in Paris, France, has published an article describing a patient with the same clinical syndrome (De novo gain-of-function variations in LYN lead to an early onset systemic autoinflammatory disorder) doi: 10.1002/art.42354. This publication also included some genetic and clinical details of patients described in the present manuscript. These patients have been presented previously at scientific meetings but limited findings, until now, have been published in abstract format only. Publication of the article by the French group will certainly affect the novelty of the findings presented here. This manuscript will need to be edited to include the newest publication and include a discussion of their findings.

Response:

After submitting our paper for review we have seen a recent case report of a 4th patient with a *LYN* missense mutation, p.Y508H, at the same amino acid (AA) position as the mutation we identified in patient 2, who has a p.Y508F mutation. The authors cite data from an abstract that we presented at a meeting that included data we shared to improve the diagnosis of patients with ultrarare diseases early, at a time when the functional evaluation of the effect of a mutation or its contribution to the clinical phenotypes is not fully established. The description of this additional patient with the same clinical phenotype validates the effect of the monogenic *LYN* mutation. Such additional cases are critical in confirming the clinical phenotype of a rare monogenic disease. **We are citing the case report and highlight the fact that the missense mutation at p.Y508H confirms the phenotype we describe that includes systemic inflammation and small vessel vasculitis, in the absence high-titer ANA and other autoantibodies. Interestingly, and consistent with our hypothesis, the missense mutation, in contrast to the truncating mutations, may confer less severe liver disease.** However, we also point out that the dysmorphic features that we describe in the girl were not seen in our patients and may be associated with additional genetic variants.

Lastly, we want to answer to the reviewer's comment regarding the novelty of the finding, which the reviewer feels are compromised by the publication of a case report that relied on the description of clinical findings described in a meeting abstract that we published before to raise awareness of this novel disease and also to guide management. It is our group's commitment to share mutations and clinical phenotypes of children with severe diseases early after discovery. The ability to test for the mutations facilitates the diagnosis of children with severe clinical phenotypes and improves their management that can save lives. In fact, all 3

additional patients were identified because the gene, *LYN*, had been included in autoinflammatory disease panels after our presentation in a rheumatology meeting, and the 3 additional mutations were found in screening for genetically characterized autoinflammatory diseases. In each of the 3 additional patients, being able to make a genetic diagnosis, helped with initiation of successful treatment and with prognosis. The pathogenic work to link that mutation to the phenotype is often long, particularly when murine models are not reflective of a disease. Sharing mutations early can sometimes result in the publication of a separate case report, as in the case mentioned. We however think that such reports are confirming the phenotype of a disease and do not compromise the pathogenic findings in our report that are out of scope of a clinical case report. We also strongly feel that publishing practices should not jeopardize the willingness of authors to share severe disease-causing mutations early prior to publication as a full manuscript as early diagnosis can help treat and manage patients with rare diseases, and even save lives.

Comment 2:

Nevertheless, the authors of the present manuscript offer much more solid evidence for the pathological effects of the novel *LYN* variants and provide some new biological insights worthwhile reporting.

However, there are some aspects which do require revision.

The authors seem to have largely neglected the autoimmune features that some of these patients have. The potential pathogenic role of B cells in the disease pathogenesis has also been minimised. The clinical features are suggestive of a condition that has overlapping autoinflammatory >>autoimmune characteristics. It is difficult to completely ignore presence of autoantibodies, even if this is transient. Unlike autoinflammatory diseases, autoimmunity might develop fully years after autoantibodies are first detected. It is uncertain if these patients will go on to develop autoimmune complication later in adulthood, particularly given the role of *Lyn* in B cell signalling and PI3K pathway activation in these cells. In my opinion, this condition might turn out to have many similarities with HA20, considering the cell-specific effects of *Lyn*, which will drive disease manifestations, some of which are probably still to be discovered. This needs to be addressed in discussion at least if no additional functional studies are possible. Therefore, although the acronym LAVLI sound catchy, in the opinion of this reviewer is that it is not entirely appropriate in this case.

Response:

We thank the reviewer for the comment and want to point out that our initial functional studies were conducted in B cells (Supplementary Figure 5). The presentation of the first patient with low serum titers of several autoantibodies (ANA, RF, anti-Sm, anti-SSA, anti-mitochondrial, anticardiolipin IgG, lupus anticoagulant) initially suggested that his disease might be autoimmune mediated and caused by immune complex deposition, B cell activation and autoimmune hepatitis. Furthermore, the murine *lyn*^{up/up} model presenting with high titer ANA and glomerulonephritis with immune complex deposition is considered a model for SLE (Hibbs

M *JEM* 2003). However, the normalization of most of the autoantibodies on low doses of steroids that did not improve the clinical features, the persistence of inflammation and of the neutrophilic rashes, and the absence of immune complex deposition on small vessel walls, as well as the absence of B cells in skin and liver biopsies led us to refocus our research towards the exploration of the role of innate immune cells and endothelial cells by using patient derived and isogenic corrected induced endothelial cells (iECs). This investigation led to the discovery of the prominent innate immune dysregulation of endothelial cells, neutrophils and monocytes as main contributors to the clinical phenotype that is consistent among our 3 patients and the recently published case report of a 3-year-old girl with a *LYN* p.Y508H mutation.

The oldest patient, (patient 2) is a young adult and so far remains ANA and ANCA negative without evidence of organ specific or systemic autoimmunity.

We have however explored the effect of the *LYN* mutation on B cell function from patients 1 and 2 in collaboration with Dr. Eric Meffre with expertise in B- cell autoimmunity. The results are described and summarized in the Supplementary Results section. **We have re-labeled the supplemental section to highlight the B cell experiments that were conducted: “Evaluation of B-cell Function and Discussion of Autoimmune Dysregulation” and have cited the data in the Results section of the main manuscript. The results of these assessments are consistent with a defect in central and peripheral B cell tolerance (Supplementary Figures i and ii, respectively).** Given the effect of the mutation in reducing central tolerance, it is more likely that the mutation is a cofactor in driving autoimmune disease in the context of additional factors including infectious challenges and additional genetic variants that converge in breaking B cell tolerance and result in autoimmune disease.

Interestingly, the autoantibody titers seen in the index patient decreased progressively on dasatinib treatment. We have therefore added the following sentence below in the supplement: “.....Anti-Sm, anti-SSA and anti-mitochondrial antibodies and lupus anticoagulant turned negative after a course of corticosteroids, and anticardiolipin IgG and rheumatoid factor turned negative after 6 months on dasatinib. ANA titers progressively decreased on corticosteroids and turned negative after 3.5 years on dasatinib therapy; it turned positive during the 11 months the patient was on etanercept monotherapy.”

We however agree with the reviewer that the possibility of the development of organ specific or systemic autoimmunity later in life cannot be ruled out and we **have therefore modified the discussion to include the need to monitor these patients for the development of features of autoimmunity as they are getting older.**

Additional comment for the reviewer: Our data suggest that the GOF mutation in *LYN* by itself drives more prominent activation of neutrophils than B cells, which may be due to the fact that *Lyn* kinase expression in mice is higher in B cells compared to neutrophils but in humans the expression level is higher in neutrophils compared to B cells (www.proteinatlas.org).

For the reasons outlined above we therefore kept the term LAVLI to characterize this disease given the clinically well-defined early onset of neutrophilic small vessel (urticarial) vasculitis and systemic inflammation that was also described in the recently published case report of a 3-year-old girl. Naming the disease does not preclude that the patients could develop autoimmune features in the future.

REVIEWERS' COMMENTS

Reviewer #1 (Remarks to the Author):

The authors have satisfied the reviewers' comments and the paper is now suitable for publication.

Reviewer #2 (Remarks to the Author):

The authors have adequately addressed Rev 2 comments. Nice work

Reviewer #3 (Remarks to the Author):

The authors have adequately addressed all my comments